

# Reconstruction of Mediterranean coastal sea level at different timescales based on tide gauge records

Jorge Ramos Alcántara[1], Damià Gomis[1], Gabriel Jordà[2]

[1]Institut Mediterrani d'Estudis Avançats (IMEDEA, University of the Balearic Islands – CSIC). Esporles (Mallorca), 07190, Spain

5    [2]Centre Oceanogràfic de Balears, C.N. Instituto Español de Oceanografía (CSIC), Palma de Mallorca, 07015, Spain

*Correspondence to*: Jorge Ramos Alcántara (jorge.ramos@uib.es)



**Abstract.** A coastal sea level reconstruction based on tide gauge observations is developed and applied to the western basin
of the Mediterranean sea. The reconstructions are carried out in four frequency bands and are based on an optimal
interpolation method in which the correlation between tide gauge data and all coastal points has been determined from the
outputs of a numerical model. The reconstructions for frequencies lower than 1 month use monthly observations from the
PSMSL database and cover the period from 1884 to 2019. For the reconstruction of higher frequencies, hourly observations
from the GESLA–2 dataset are used, and cover from 1980 to 2015. Total sea level is retrieved with high accuracy from the
merging of the different frequency bands. Results of a cross–validation test show that independent tide gauge series are
highly correlated with the reconstructions. Moreover, they correlate significantly better with the reconstructions than with
altimetry data in all frequency bands, and therefore the reconstruction represents a valuable contribution to the attempts of
recovering coastal sea level. The obtained reconstructions allow the characterization of the coastal sea level variability, to
estimate coastal sea level trends and to examine the correlation between Western Mediterranean coastal sea level and the
main North Atlantic climate indices. The limitations and applicability of the method to other regions are also discussed.

## 1 Introduction

The coastal zone is a fragile region exposed to sea level variations at different time scales. On the one hand, climate
change–induced mean sea level rise is expected to have a major impact on low elevation coasts. These are mainly socio–
economic impacts (forcing millions of people to move inland, or to develop costly coastal protections), although the
ecological impacts associated with the alteration of the sediment budget in coastal regions could also be important
(FitzGerald et al., 2008; Kirwan et al., 2010). On the other hand, rapid variations in sea level associated with extreme events
(tsunamis, meteotsunamis or storm surges) can also have devastating effects in coastal regions. Climate models project
increases in the frequency of some extreme events, which will in any case intensify their impacts as a result of rising mean
sea level (Spalding et al., 2014).

In the particular case of southern Europe, a large part of the economy depends on coastal activities. Therefore, it is
expected that mean sea–level rise and its associated hazards will have significant impacts on the Mediterranean coasts
(Wolff et al., 2018). These impacts include coastal erosion, flooding, damage to coastal structures, or saline intrusion in
estuaries and aquifers (Jordà et al., 2012; le Cozannet et al., 2017). Sea level changes (whether gradual mean sea level rise or
changes in extreme events) and their impacts are not uniform in space, which makes necessary to study their variability at
regional scale, and not only at global scale (Lyu et al., 2014). The processes that introduce small–scale variability in sea level
are diverse, but they are particularly relevant in coastal areas due to their shallow depth and to the complexity of topography
and bathymetry. For example, the continental slope largely decouples the dynamics of the open ocean from that of the
coastal region (Woodworth et al., 2019).

In order to carry out a proper coastal management (which nowadays include adaptation strategies to climate change) it is
compulsory to have oceanographic databases that allow the understanding of the of the spatial and temporal sea level



variability. In addition to global scale sea level drivers such as the increase in the amount of water in the oceans due to continental ice melting or the thermal expansion, at regional scale there are other key drivers such as the meteorological component (forcing of atmospheric pressure and wind; Gomis et al., 2012) or the coastal circulation. These have a spatio–temporal variability that is not always captured by the current observational networks without some further data processing.

The study of extreme sea levels is also particularly important due to the impact of these events. Extreme sea levels are caused by different processes and forcings, some of which may vary in intensity and frequency over time (Tsimplis and Shaw, 2010).

A first source of sea level observations are tide gauges, which cover different time periods (few records are available before 1960, but there are also series dating back to the 17th century). Whereas tide gauges generally provide very accurate

measurements (Douglas, 2001), their main limitation is that they are point–wise measurements with a heterogeneous spatial and temporal distribution. Furthermore, for climate studies it must be taken into account that their measurements are affected by the vertical motion of the ground where they are anchored (Cipollini et al., 2017), which makes necessary to have accurate local estimates of vertical land motion in order to isolate the marine contribution of tide gauge records (Marcos et al., 2019).

Since 1992, sea level measurements provided by satellite altimetry are also available. This technique has a global coverage, and by minimising all sources of error affecting the measurements, accuracy close to 1 cm can be achieved (Cazenave et al., 2018). However, altimetric measurements in coastal regions are particularly complex; despite the advances reached in recent years, standard altimetric data are only available from 5–10 km offshore (Marcos et al., 2019; Vignudelli et al., 2019). Altimetric products have also limited spatial and temporal resolution: the separation between adjacent satellite

ground tracks is between 50 and 300 km, and the revisiting time is between a few days and a few weeks (Marcos et al., 2019).

In addition to the observations described above, in the Mediterranean region there are also some sea level reconstructions based on different data sources that have allowed to deepen the understanding of sea level variability in the region. Thus, Holgate and Woodworth (2004) produced direct estimations of regional trends by averaging tide gauge series. Others (e.g.

Tsimplis et al., 2008; Calafat and Gomis, 2009; Meyssignac et al., 2011) combined data from tide gauges, altimetry and numerical model outputs following the reduced space optimal interpolation methodology proposed by Kaplan et al., (1997). However, all these reconstructions focused on the variability of the open ocean and they have neither the spatial resolution nor the temporal resolution required to characterise coastal processes.

This paper presents a sea level reconstruction for the Western Mediterranean coasts that meets two fundamental

requirements: i) it covers all coastal regions; and ii) it has the spatial and temporal resolution required to characterise coastal processes. A cross–validation test will demonstrate that it provides better estimates than coastal altimetry, and therefore that it represents a valuable contribution to the attempts of recovering coastal sea level. Carrying out the reconstruction for different frequency bands will allow to deepen our knowledge on sea level variability at different time scales down to a daily scale (that is, beyond the temporal resolution of previous reconstructions). The methodology followed to obtain the coastal



sea level reconstruction is based on the optimal interpolation method; this methodology also provides error estimations, which is essential in this type of reconstructions.

The paper is organized as follows: first, Sect. 2 overviews the different datasets used along this work. Section 3 reviews the optimal interpolation method, detailing how it has been adapted to reconstruct sea level in different frequency bands; the different validation methods used to evaluate the accuracy of the reconstructions are also briefly described. In Sect. 4 we

present the main results: the reconstructions, both for different frequency bands and for total sea level, altogether with the results of the cross–validation test and an estimate of the interpolation errors. In Sect. 5 the reconstructions are used to infer some aspects of coastal level variability in the western Mediterranean. All results are discussed in Sect. 6, examining the limitations of the method and comparing the reconstructions with coastal altimetry products and open ocean reconstructions. Finally, conclusions are presented in Sect. 7.

## 85   **2 The data sets**

### **2.1 Tide gauge data**

We have used two tide gauge datasets: GESLA–2 (Global Extreme Sea Level Analysis, https://www.gesla.org/), which has data with a maximum frequency of 1 hour, and PSMSL (Permanent Service for Mean Sea Level, https://www.psmsl.org/), with monthly data. The GESLA database was created from the need to have information on

extreme events (in particular on their interannual variability), and in a first version (GESLA–1, which was not published) it collected data from two international banks: UHSLC (University of Hawaii Sea Level Center) and GLOSS 2 (Global Sea Level Observing System), as well as from several national banks. In 2016, with the aim of updating the database and extending its spatial coverage, the GESLA–2 set was published. In this new set, the geographical coverage of some regions (including the Mediterranean Sea) was improved, especially for the second half of the 20th century. GESLA–2 has allowed

more globally representative analyses since about 1970 (Woodworth et al., 2016). GESLA–2 has its own quality control: the quality and possible use of each data is specified (Piccioni et al., 2019). In this work, only the measurements marked as "correct" were selected. For the period between 1980 and 2015 and for the western Mediterranean basin, 34 tide gauge records are available (Table 1). Some of the series showed obvious datum shifts; this made necessary to first identify the intervals with different datums and subtract their means, in order to convert the original data into zero–mean anomalies.

The PSMSL, managed by the National Oceanography Centre in Liverpool, collects data from numerous institutions around the world. This bank has Revised Local Reference (RLR) series, which are referenced to a common datum, and constitute approximately two thirds of the total number of stations. All the series used in this work are of the RLR type. Although the PSMSL has some series starting in the late 19th century, the number of stations increases considerably in the second half of the 20th century, concentrating along the most developed coastal regions, especially Europe and North

Americ (Holgate et al., 2013). From the PSMSL, 38 tide gauges were selected from the western Mediterranean basin (Table



2). In this case the reconstruction was carried out from the time when the first tide gauge has monthly measurements, namely the Genoa tide gauge, whose series starts in 1884.

The glacial isostatic adjustment (GIA) has not been applied to the tide gauge series. This correction is rather small in the Mediterranean, except for the Adriatic Sea (Marcos and Tsimplis, 2007a), which is not part of our reconstruction domain.


**Table 1: GESLA–2 Tide Gauges in the Western Mediterranean Basin, with their Locations, Start and End Dates of the Series, and the Percentage of Missing Data.**

| Station | Latitude | Longitude | First data | Last data | Missing values (%) |
|---|---|---|---|---|---|
| Ajaccio | 41.92 | 8.76 | 1981-07-25 | 2014-09-22 | 58.64 |
| Alcudia | 39.83 | 3.14 | 2009-09-11 | 2014-12-31 | 0.52 |
| Algeciras | 36.12 | -5.43 | 1980-01-01 | 2012-12-31 | 16.43 |
| Almería | 36.83 | -2.48 | 2006-01-01 | 2014-12-31 | 2.13 |
| Barcelona | 41.34 | 2.16 | 1993-01-01 | 2014-12-31 | 2.05 |
| Cagliari | 39.21 | 9.11 | 1986-12-17 | 2010-11-30 | 19.60 |
| Carloforte | 39.15 | 8.31 | 1988-06-27 | 2010-12-01 | 26.91 |
| Ceuta | 35.90 | -5.32 | 1980-01-01 | 2012-12-31 | 5.53 |
| Formentera | 38.73 | 1.42 | 2009-09-25 | 2014-12-31 | 1.77 |
| Fos–sur–Mer | 43.40 | 4.89 | 2006-01-31 | 2014-09-22 | 50.17 |
| Gandía | 39.00 | -0.15 | 2007-07-06 | 2014-12-31 | 1.46 |
| Genova | 44.41 | 8.93 | 1998-08-06 | 2010-10-13 | 2.07 |
| Gibraltar | 36.12 | -5.35 | 1980-01-01 | 2000-04-30 | 33.45 |
| Ibiza | 38.91 | 1.45 | 2003-01-01 | 2014-12-31 | 1.14 |
| Imperia | 43.88 | 8.02 | 1986-12-11 | 2010-10-13 | 30.60 |
| La Figueirette | 43.48 | 6.93 | 2011-05-24 | 2014-12-31 | 2.43 |
| Mahón | 39.89 | 4.27 | 2009-10-29 | 2014-12-31 | 0.21 |
| Málaga | 36.72 | -4.42 | 1980-01-01 | 2014-12-31 | 0.38 |
| Marseille | 43.30 | 5.35 | 1985-01-01 | 2014-12-31 | 45.97 |
| Melilla | 35.29 | -2.92 | 2007-10-23 | 2014-12-31 | 1.07 |
| Monaco Fontvieille | 43.73 | 7.42 | 1980-12-31 | 2014-12-31 | 46.71 |
| Monaco Port Hercule | 43.73 | 7.42 | 1999-04-15 | 2010-12-01 | 1.98 |
| Motril | 36.72 | -3.52 | 2005-01-01 | 2014-12-31 | 0.25 |
| Nice | 43.70 | 7.29 | 1981-07-03 | 2014-12-31 | 47.45 |
| Palma de Mallorca | 39.56 | 2.64 | 2009-09-11 | 2014-12-31 | 0.15 |
| Port Camargue | 43.52 | 4.13 | 2009-11-05 | 2012-12-31 | 20.56 |
| Port Ferreol | 43.36 | 6.72 | 2012-03-29 | 2014-12-31 | 1.39 |
| Port Vendres | 42.52 | 3.11 | 1981-12-28 | 2014-04-02 | 29.21 |
| Porto Torres | 40.84 | 8.40 | 1985-05-22 | 2010-11-30 | 34.73 |
| Sagunto | 39.63 | -0.21 | 2007-06-13 | 2014-12-31 | 0.54 |
| Sète | 43.40 | 3.70 | 1992-01-07 | 2014-02-07 | 19.71 |
| Tarifa | 36.00 | -5.60 | 1980-01-01 | 2014-12-31 | 7.15 |



| Station | Latitude | Longitude | First data | Last data | Missing values (%) |
|---|---|---|---|---|---|
| Toulon | 43.12 | 5.91 | 1981-06-28 | 2014-09-22 | 33.37 |
| Valencia | 39.46 | -0.33 | 1992-10-01 | 2014-12-31 | 1.46 |

**Table 2: PSMSL Tide Gauges in the Western Mediterranean Basin, with their Locations, Start and End Dates of the Series, and the Percentage of Missing Data.**

| Station | Latitude | Longitude | First data | Last data | Missing values (%) |
|---|---|---|---|---|---|
| Ajaccio | 41.92 | 8.76 | 1981-08-15 | 2019-06-15 | 50.33 |
| Alcudia | 39.83 | 3.14 | 2009-10-15 | 2018-12-15 | 1.80 |
| Algeciras | 36.12 | -5.43 | 1943-07-15 | 2018-12-15 | 19.54 |
| Alicante | 38.34 | -0.48 | 1960-01-15 | 1997-12-15 | 3.07 |
| Almería | 36.83 | -2.48 | 1977-11-15 | 2018-12-15 | 24.09 |
| Barcelona | 41.34 | 2.17 | 1993-01-15 | 2018-12-15 | 3.85 |
| Cagliari | 39.20 | 9.17 | 15/08/1896 | 2014-12-15 | 57.71 |
| Carloforte | 39.15 | 8.31 | 2001-01-15 | 2015-12-15 | 0.00 |
| Cartagena | 37.60 | -0.97 | 1977-05-15 | 1987-11-15 | 12.60 |
| Ceuta | 35.89 | -5.32 | 1944-03-15 | 2018-12-15 | 3.23 |
| Formentera | 38.73 | 1.42 | 2009-10-15 | 2018-12-15 | 8.11 |
| Fos–sur–Mer | 43.40 | 4.89 | 2006-02-15 | 2019-06-15 | 34.16 |
| Gandía | 39.00 | -0.15 | 2007-07-15 | 2018-12-15 | 1.45 |
| Genova | 44.40 | 8.90 | 15/06/1884 | 2014-12-15 | 21.57 |
| Gibraltar | 36.15 | -5.36 | 1961-07-15 | 2014-05-15 | 39.84 |
| Ibiza | 38.91 | 1.45 | 2003-02-15 | 2018-12-15 | 1.05 |
| Imperia | 43.88 | 8.02 | 2001-01-15 | 2015-12-15 | 0.00 |
| L'Estartit | 42.05 | 3.21 | 1990-01-15 | 2019-06-15 | 0.00 |
| Mahón | 39.89 | 4.27 | 2009-11-15 | 2018-12-15 | 2.73 |
| Málaga | 36.71 | -4.42 | 1944-01-15 | 2018-12-15 | 16.44 |
| Marseille | 43.28 | 5.35 | 15/02/1885 | 2019-06-15 | 6.20 |
| Melilla | 35.29 | -2.93 | 2008-01-15 | 2018-12-15 | 5.30 |
| Monaco | 43.73 | 7.42 | 1956-01-15 | 2019-06-15 | 48.16 |
| Motril | 36.72 | -3.52 | 2005-01-15 | 2018-12-15 | 1.19 |
| Nice | 43.70 | 7.29 | 1978-01-15 | 2019-06-15 | 12.65 |
| Palma de Mallorca | 39.55 | 2.62 | 1964-01-15 | 2018-12-15 | 60.76 |
| Port Ferreol | 43.36 | 6.72 | 2012-04-15 | 2019-06-15 | 6.90 |
| Port–la–Nouvelle | 43.01 | 3.06 | 2013-06-15 | 2019-06-15 | 1.37 |
| Port–Vendres | 42.52 | 3.11 | 1984-01-15 | 2019-06-15 | 24.18 |
| Porto Maurizio | 43.87 | 8.02 | 15/08/1896 | 1922-07-15 | 0.96 |
| Porto Torres | 40.84 | 8.40 | 2001-01-15 | 2015-12-15 | 2.22 |
| Sagunto | 39.63 | -0.21 | 2007-09-15 | 2018-12-15 | 2.94 |
| Sète | 43.40 | 3.70 | 1992-01-15 | 2019-06-15 | 14.85 |
| Tarifa | 36.01 | -5.60 | 1943-09-15 | 2018-12-15 | 5.64 |
| Tarragona | 41.08 | 1.21 | 2011-06-15 | 2018-12-15 | 1.10 |





| Toulon | 43.11 | 5.91 | 1961-01-15 | 2019-06-15 | 43.45 |
| Valencia | 39.44 | -0.31 | 1994-01-15 | 2018-12-15 | 2.00 |
| Villa Sanjurjo | 35.25 | -3.92 | 1944-01-15 | 1949-11-15 | 0.00 |

## 2.2 SOCIB WMOP model

In order to obtain information on the dynamic relationships between different locations (see Sect. 3), the outputs of the numerical model managed by the Balearic Islands Coastal Observing and Forecasting System (SOCIB) have been used.
SOCIB is a multi–platform observatory whose products include numerical model outputs to support operational oceanography. Through a regional configuration of the ROMS model (Shchepetkin and McWilliams, 2005) for the western Mediterranean basin, SOCIB has implemented a forecasting system called WMOP that provides daily forecasts with a 3 days time horizon for temperature, salinity, sea level and currents (Juza et al., 2016). The daily forecasts as well as all model outputs from the moment it was implemented can be downloaded from the SOCIB website (https://www.socib.es). For sea
level, the spatial resolution of the outputs varies between 1.8 and 2.2 km and they are available every 3–4 hours since August 2013 (Juza et al., 2016). The coastal points of this model grid were used to define the grid for the sea level reconstructions presented later on, as the spatial correlations on which the optimal interpolation method is based were calculated from the model outputs.

## 2.3 Altimetry data and dynamic atmospheric correction

The European observation programme Copernicus provides, through the Copernicus Marine Environment Monitoring Service (CMEMS), regular and systematic baseline information on the global ocean and European seas. Among others CMEMS delivers sea level products derived from a direct processing of altimetric observations (von Schuckmann et al., 2018). In order to compare the coastal reconstructions obtained in this work with the latest generation of altimetric data, sea level anomalies from multi-mission satellite altimetry products were downloaded from the CMEMS website
(https://resources.marine.copernicus.eu/?option=com_csw&view=details&product_id=-SEALEVEL_MED_PHY_L4_REP_OBSERVATIONS_008_051). The processing of this dataset includes some corrections such as the removal of high–frequency variability, implemented to avoid the aliasing that could result from the low spatio–temporal resolution of altimetric observations (Gomis et al., 2012). This correction, called DAC (Dynamic Atmospheric Correction), removes a significant part of the sea level variability associated with the atmospheric component, and therefore
had to be reversed. The DAC is produced by CLS using the Mog2D model from Legos (Carrère and Lyard, 2003) and distributed by Aviso+, with support from CNES (https://www.aviso.altimetry.fr/en/data/products/auxiliary-products/dynamic-atmospheric-correction/description-atmospheric-corrections.html). These DAC data were bilinearly interpolated onto the coastal points where altimetric sea level anomalies were available and added to these series.



### 2.4 Climatic indices

In order to study the relationship between the variability inferred from our coastal sea level reconstructions and large–scale atmospheric modes, climatic indices were downloaded for the four modes that are most relevant for the Western Mediterranean basin: the North Atlantic Oscillation (NAO), the East Atlantic pattern (EA), the East Atlantic/Western Russian (EA/WR) and the Scandinavian pattern (SCAN). The NAO index is the main mode of variability in winter, and accounts for the large–scale variation in atmospheric mass between the areas of the Azores subtropical anticyclone and the low–pressure area near Iceland. The EA index influences the freshwater flux into the north–east Atlantic, and also influences the western Mediterranean basin. The EA/WR index contributes to heat fluxes and has an impact on precipitation in the Mediterranean. The SCAN index is related to the climate of the Scandinavian Peninsula and East Asia, as well as to the precipitation in northern Europe (Bueh and Nakamura, 2007; Josey et al., 2011; Martínez-Asensio et al., 2014). All series were obtained from the NOAA Climate Prediction Centre (http://www.cpc.ncep.noaa.gov/data/teledoc/telecontents.shtml) and consist of monthly data covering the period from 1950 to present.

### 3 Methodology

#### 3.1 Optimal Interpolation

Optimal Interpolation is a type of linear statistical interpolation in which the best representation of the state vector of a system at a given point ($\phi_g$) is obtained through the superposition of a first guess at that point ($\phi_g^{fg}$) and the weighting of the differences between observed data ($\phi_i^o$) and those estimated by the first guess at each observation point ($\phi_i^{fg}$) (Hasselmann et al., 1997):

$$\phi_g = \phi_g{}^{fg} + W_{gi}(\phi_i{}^o - \phi_i{}^{fg}) = \phi_g{}^{fg} + W_{gi}\,\phi_i{}' \tag{1a}$$

where $W_{gi}$ are the weights applied to the differences at each observation point $i$ to obtain the state vector at point $g$. For a discrete set of $m$ interpolation points, Eq. (1a) can be written as:

$$\underline{\phi_g} = \underline{\phi_g}{}^{fg} + \underline{\underline{W}} \cdot \underline{\phi}' \tag{1b}$$

where $\underline{\phi_g}$ and $\underline{\phi_g}{}^{fg}$ are $m$-vectors, $\underline{\phi}'$ is the $n$-vector of anomalies at the $n$ observation points, and $\underline{\underline{W}}$ is the $m$ x $n$ weight matrix. The method is named as 'Optimal Interpolation' because the weights are determined through the statistical minimisation of the mean square error between the real and the interpolated field. The development of this minimisation leads to the expression:

$$\underline{\phi_g} = \underline{\phi_g}{}^{fg} + \underline{\underline{\theta}}\,\underline{\underline{T}}^{-1}\,\underline{\phi}' \tag{2}$$



where $\boldsymbol{\theta}$ is an $m$ x $n$ matrix whose elements are correlation values between series at interpolation points and at observation points, and $\boldsymbol{T}$ is an $n$ x $n$ matrix whose elements are correlation values between series at observation points. Under the assumption of spatially uncorrelated noise, this has no effect on $\boldsymbol{\theta}$, but the diagonal of $\boldsymbol{T}$ (the correlation of observation series with themselves) includes the noise of the observations. Thus, matrix $\boldsymbol{T}$ can be expressed in terms of a correlation matrix between true values $\boldsymbol{\theta}^o$, the noise–to–signal coefficient (variance of the errors divided by the variance of

the signal) of the observations $\eta^2$ and the identity matrix $\boldsymbol{I}$:

$$\boldsymbol{T} = \boldsymbol{\theta}^o + \eta^2 \boldsymbol{I} \qquad (3)$$

where the noise–to–signal coefficient $\eta^2$ has been assumed to be the same for all observation points. This method also provides an explicit expression of the interpolation error in a statistical sense. That is, the mean value of the interpolation errors that would result if an infinite number of realizations of the observed field were interpolated in the same way (with the

same observation points and the same weights). The errors at each interpolation point ($\varepsilon_g$) is given by:

$$\varepsilon_g = \sigma_g \left( 1 - Tr_g[\boldsymbol{\theta}^T \boldsymbol{T}^{-1} \boldsymbol{\theta}] \right) \qquad (4)$$

where $\sigma_g$ is the variance of the signal at point $g$, and $Tr_g[\bullet]$ denotes the trace element of matrix $[\bullet]$ corresponding to the interpolation point $g$.

## 3.2 Implementation and evaluation of the reconstruction methodology

The reconstruction has been performed in different frequency bands. The main reason is that spatial correlations may differ for different time scales (e.g., daily variability may have associated shorter spatial scales than multidecadal changes). Thus, the splitting in frequency bands allows a more accurate spatial interpolation. Furthermore, because high–frequency signals usually dominate and mask low frequencies, the separation in frequency bands allows a better representation of low frequencies (e.g., interannual and decadal variability).

The monthly PSMSL data were separated into three frequency bands: a first one corresponding to periods longer than 10 years (T>10y), a second one corresponding to periods between 1 and 10 years (1y<T<10y), and a third one corresponding to periods between 1 month and 1 year (1m<T<1y). The GESLA–2 hourly data were first averaged into daily data, and then a frequency band corresponding to periods between 1day and 1month (1d<T<1m) was isolated. The number of stations considered for each frequency band was different (Fig.1) and dependent on the length of the series. Namely:

• For the T>10y band, 5 series with at least 20 years of consecutive data were selected. The frequency band was isolated by means of a low–pass filtering carried out using a Butterworth filter of order 10, subtracting from each series the average of a period in which all of them had data.




- For the band 1y<T<10y, tide gauges with at least 10 years of consecutive data were considered. First, the frequency band isolated in the previous point was subtracted from the original series and then the frequencies corresponding to periods T<1y were removed, also by means of a Butterworth filter of order 10.

- For the band 1m<T<1y, all available PSMSL tide gauges were considered, and the two frequency bands isolated in the previous points were removed from the original series. As these consisted of monthly data, there was no need to remove the periods T<1m.

- For the 1d<T<1m band, the three previous bands (obtained from PSMSL data) were subtracted from each of the GESLA–2 series (this required a prior conversion of the three bands to daily values by means of linear interpolation).

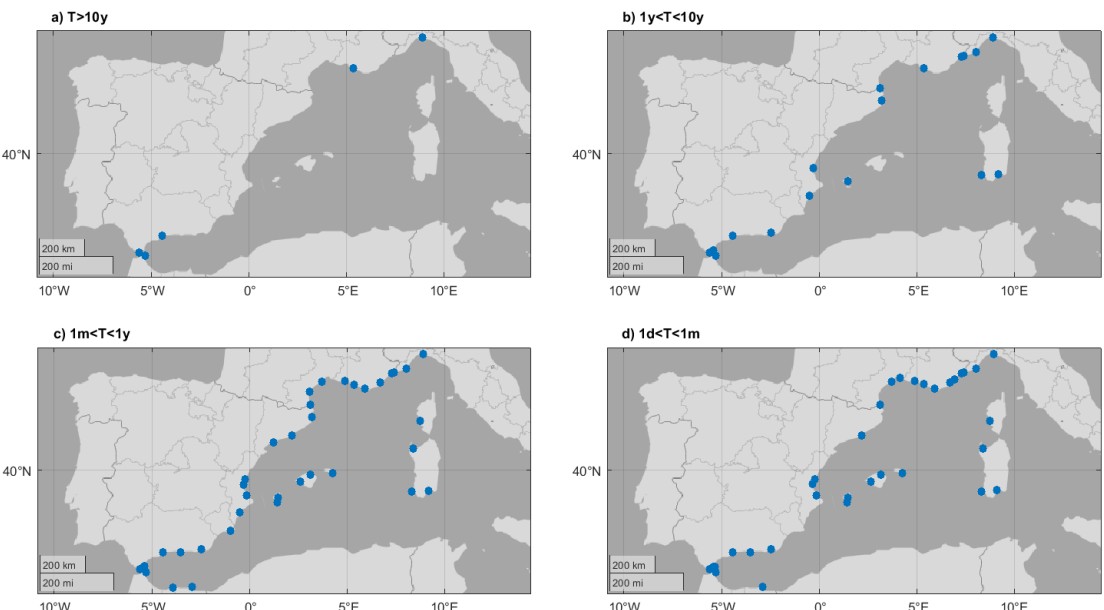

**Figure 1: Maps of the tide gauges selected in each band: (a) PSMSL tide gauges used in the reconstruction for the frequency band below 10 years. (b) PSMSL tide gauges used in the reconstruction for the frequency band between 1 and 10 years. (c) PSMSL tide gauges used in the reconstruction for the frequency band between 1 month and 1 year. (d) GESLA–2 tide gauges used in the reconstruction for the frequency band between 1 day and 1 month.**

The implementation of the optimal interpolation required to estimate the correlations between interpolation points and observation points, and also between each pair of observation points. For this purpose, two approaches were followed, one based on model data (used to interpolate the frequency bands 1y<T<10y, 1m<T<1y and 1d<T<1m) and a second one based on fitting an analytical correlation function (used to interpolate the band T>10y, for which the period spanned by the model does not allow a reliable estimation of correlations). These two approaches are described in the following.

- Calculating correlations from numerical model outputs: defining the interpolation points as the coastal points of the SOCIB model grid and approximating the location of each tide gauge to the nearest grid point (which implies a



minimum error given the spatial resolution of the model) makes the calculation of all necessary correlations
straightforward. The elements of matrices $\boldsymbol{\theta}$ and $\boldsymbol{\theta}^o$ appearing in Eq. (2) and Eq. (3), which correspond to correlations
between true values of the field, were computed in this way for the three frequency bands.

In order to validate this procedure, the correlations obtained for the pairs of SOCIB model series collocated with tide
gauges were compared with those obtained for the original tide gauge series. The latter were found to be lower than the
correlations between model series, due to the presence of observational noise. In order to simulate observations with
model data, a Gaussian noise was added to the model series closest to the tide gauges, with a variance adjusted to make
model correlations as close as possible to tide gauge correlations. These model series with added noise were used as
pseudo–observations to carry out a first test: the optimal interpolation of these pseudo–observations at all coastal points
was compared with the original model series with the aim of verifying the ability of the method to reproduce coastal sea
level at all locations from a discrete number of observations. For more information on this validation test, Appendix A.

• By fitting an analytical correlation function: correlation functions typically depend inversely on distance (e.g., Gaussian
functions; Rasmussen, 1996). For the 5 tide gauges considered for the frequency band T>10y, the following function
was used:

$$\theta_{ij} = e^{-\frac{d_{ij}}{2L_S{}^2}} \cdot e^{-\frac{dt}{L_t}} \tag{5}$$

where $d_{ij}$ represents the distance between locations $i$ and $j$, and $L_S$ is the characteristic length scale of spatial
correlation. This was fitted considering the correlations between the 11 tide gauges available in the whole Mediterranean
sea with long enough time series, resulting in a value of 1254 km. The second part of the expression corresponds to a
temporal correlation between observations and is not always used in Optimal Interpolation. In our case, using
observations from different times intends to compensate for the small number of observations available at a given time;
namely, we considered observations two years ahead and two years behind the time of each interpolated value,
obviously with weights decreasing with the time distance, $dt$. The characteristic scale of temporal correlations, $L_t$, was
set to two years.

Another parameter necessary for the implementation of Optimal Interpolation is the noise–to–signal coefficient of
observations appearing in Eq. (3). This coefficient was optimised using the golden section search, which is appropriate for
finding the minimum or maximum of unimodal functions through successive reduction of the range of values within which
the extreme is known to exist (Pejic and Arsic, 2019). In our case, we searched for the noise–to–signal coefficient that
minimised the mean square error between the original tide gauge series and the reconstructed series through a cross–
validation test that will be explained in the next section.



### 3.3 Validation of the reconstructions

In order to make a diagnosis of the reconstructions, cross–validation tests were carried out. These tests use part of the available observations to fit the model, while the other part is used as a validation set (Hastie et al., 2008).  In our case, the sea level series at the closest grid points to each tide gauge were obtained considering in each case as observations all available tide gauges except the one closest to that grid point, which was kept as validation series. The reconstructed series were then compared with the tide gauge series using several statistics, namely: i) the root mean square error (RMSE),

considered as a standard metric for model errors; ii) the percentage of the variance of observations explained by the reconstruction; iii) the Pearson correlation coefficient between the reconstructed series and the tide gauge series. This was done for each frequency band in which the reconstructions were carried out.

Another validation test consisted of recovering the original monthly signals from the sum of the first three frequency band reconstructions, and of the original daily signals from the sum of the four band reconstructions. The comparison of these

unified signals with the original tide gauge series allows to verify that the split into frequency bands has been carried out correctly, and that it is possible to reconstruct the complete series at all interpolation points. From the interpolation errors calculated for the reconstructions of the different frequency bands, the theoretical interpolation error of the total reconstruction can also be obtained: assuming that the errors of the reconstructions in the different frequency bands are independent of each other, the variance of the total error is the quadratic sum of the variances of the error in each band.

Finally, the total reconstructed series, as well as the reconstructed series of each frequency band, were compared with the last generation altimetric products and checked against the original tide gauge series, in order to determine the goodness of each approximation.

### 4 Coastal Reconstruction Validation

### 4.1 Results of the cross–validation test

In general, the statistics of the cross–validation test described in Sect. 3.3 gave good results when applied to the reconstructions of the 4 frequency bands (Fig. 2, 3 and 4): most reconstructions explain a high percentage of the variance of the original series and also show good correlations with the original series. The lowest frequency band (T>10y) displays the best results: RMS differences are below 2 cm for all tide gauges, and correlations range from a minimum value of 0.74 in Ceuta and a maximum value of 0.99 in Marseille. For this band, the reconstruction explains more than 75% of the variance

of the original series at all stations except in Ceuta, where it only explain 39%. These results show that a few stations (only 4 in the present case) are enough to reconstruct the low–frequency (decadal) variability, since this is mainly associated with large–scale spatial structures (Woodworth et al., 2019). Considering the observations from the two years before and after each interpolation time has also helped to obtain good estimates.

For the interannual to interdecadal frequency band (1y<T<10y) the reconstructions explain in general smaller percentages

of the tide gauge variance than for the other frequency bands. The best results are obtained at the stations located on the east





coast of the Iberian Peninsula, southern France and northern Italy, where the explained variance is at least 50% (>75% for the Imperia, L'Estartit and Port Vendres tide gauges), RMS differences are below 4 cm and correlations are higher than 0.7. However, for the stations close to the Strait of Gibraltar the statistics are much poorer, e.g., the reconstructions can explain almost none of the variance of the original series. The results for Ibiza and Cagliari are not good either. In order to check

whether the origin of the poor statistics of this band was due to a bad representation of the correlation matrix elements used for the Optimal Interpolation, we also tested analytical correlation functions with different correlation characteristic lengths; however, no improvement in the results was achieved. This suggests that the stations showing poor statistics would have a sea level variability spatially decoupled from the others at this frequency band and/or there were problems with the observations.

In the intra–annual frequency band (1m<T<1y) the statistics are better than for the previous band, with the worst values corresponding to the same stations: those located near the strait of Gibraltar, Ibiza and Cagliari. For the other stations, RMS differences are below 3 cm, correlations above 0.8, and explained variances above 70 % in almost all cases.

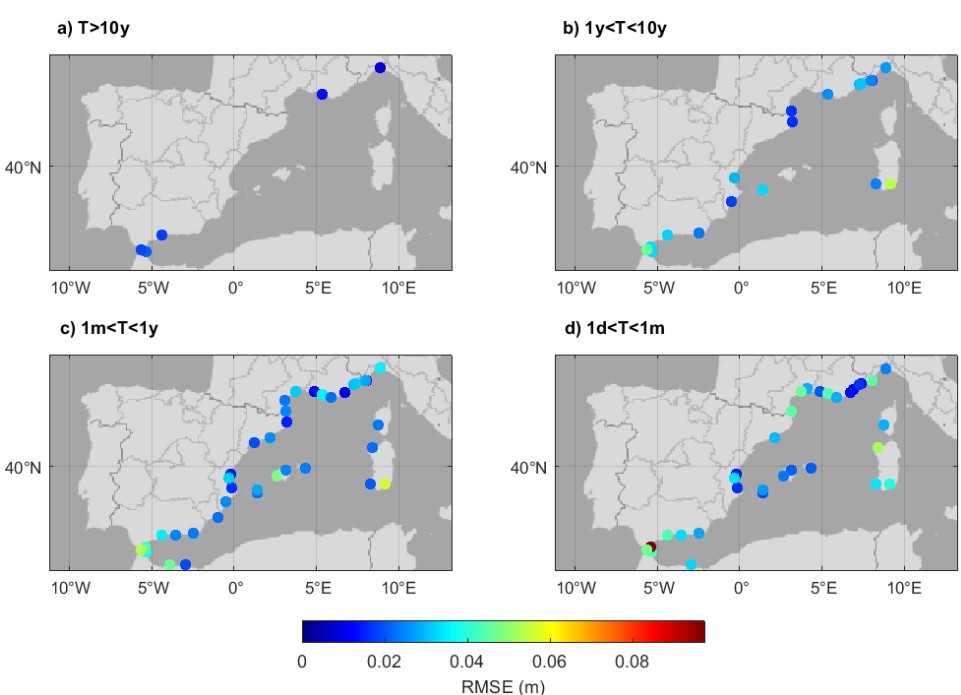

**Figure 2: Results of the cross–validation test (I): RMS differences between the reconstructed values and the original series for the four frequency bands: a) T>10y, b) 1y>T>T>10y, c) 1m>T>1y, d) 1d>T>1m.**





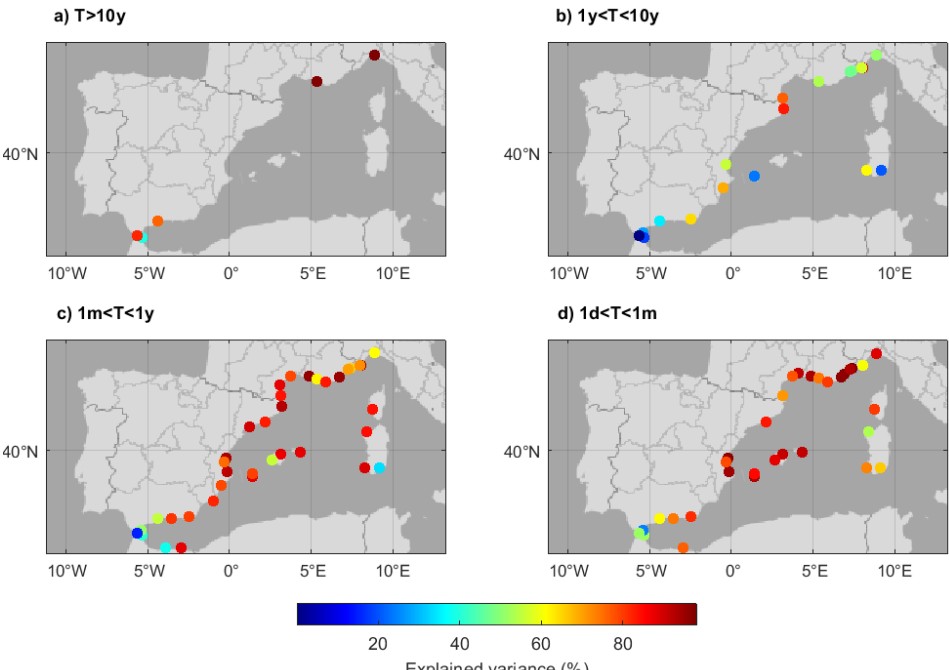

**Figure 3: Results of the cross–validation test (II): percentage of variance of the original tide gauge series explained by the reconstructions for the four frequency bands: a) T>10y, b) 1y>T>T>10y, c) 1m>T>1y, d) 1d>T>1m.**

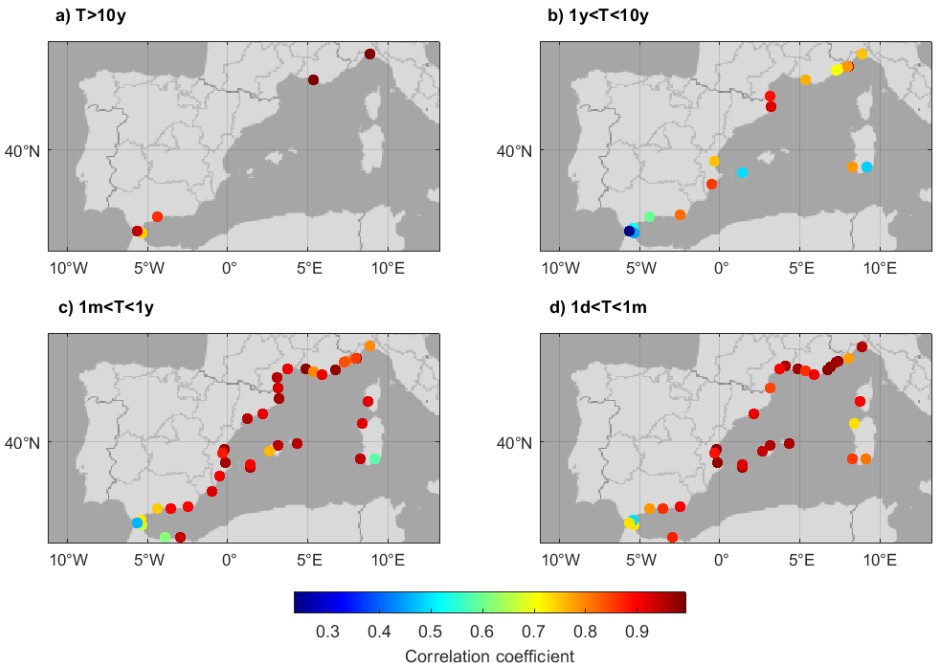

**Figure 4: Results of the cross–validation test (III): correlation values between the reconstructed and the original tide gauge series for the four frequency bands: a) T>10y, b) 1y>T>T>10y, c) 1m>T>1y, d) 1d>T>1m.**




Finally, for the daily to monthly frequency band the statistics is also good, with RMS differences below 5 cm, correlations above 0.75, and explained variances above 60 % in almost all cases. The exceptions are Porto Torres and, again, the stations located near the strait of Gibraltar, where RMS differences almost reach 10 cm and the reconstruction explains less than 25% of the original variance.

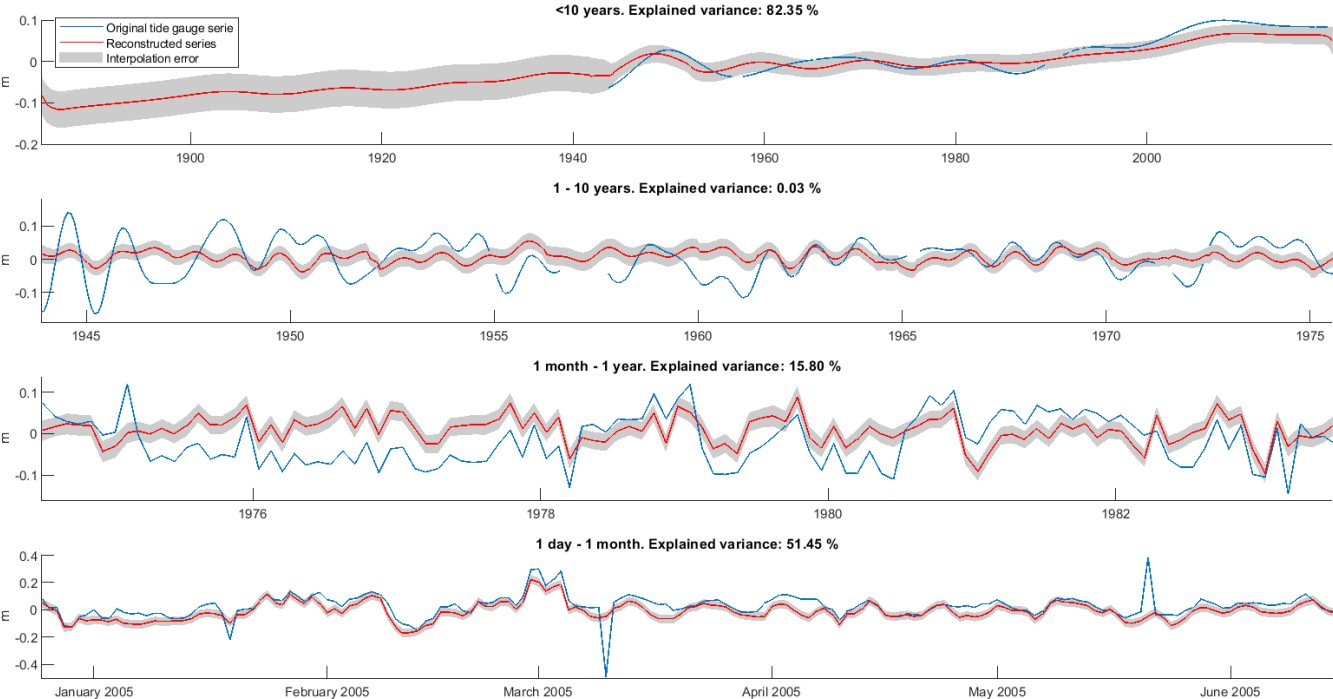

**Figure 5: Comparison between the Tarifa tide gauge series and its reconstruction (for the four frequency bands), as given by the cross–validation test. The statistical interpolation error given by the Optimal Interpolation formulation (eq. (4)) is plotted in the form of an uncertainty for the interpolated values. Different time axes have been used for the different frequency bands, in order to correctly appreciate the variability of each band.**

In order to visualize the bad results given by the cross–validation test near the strait of Gibraltar, we show a comparison between the original Tarifa tide gauge series and its reconstructions given by the cross–validation test for the four frequency

bands (Fig. 5). Tarifa station was chosen because its series has been used for the reconstruction of all four frequency bands. The difficulties to reconstruct the original tide gauge series for the intra–annual and inter–annual frequency bands are well apparent. For these frequency bands, the differences between the original and the reconstructed series are larger than the statistical interpolation error given by the Optimal Interpolation formulation (Eq. (4)); this suggests that for some stations the correlation elements of the Optimal Interpolation matrices are not correctly represented. The plots also suggest that in some

frequency bands the observations may have some problems that were not identified by the quality control. For instance, in the 1 day–1 month the explained variances are lowered by the spikes in the observations, which look unrealistic. Besides For




comparison, Fig. 6 shows the same as Fig. 5 but for the Genoa station, which generally shows good statistics. As for Fig. 5, results are worse for the inter–annual and intra–annual bands (though notably better than in the case of Tarifa). This suggests that some of the coastal processes driving these frequency bands cannot be correctly interpolated by the reconstruction 290 method. On the other hand, in the daily to monthly frequency band the reconstruction is able to explain more than 88 % of the variance of the original series.

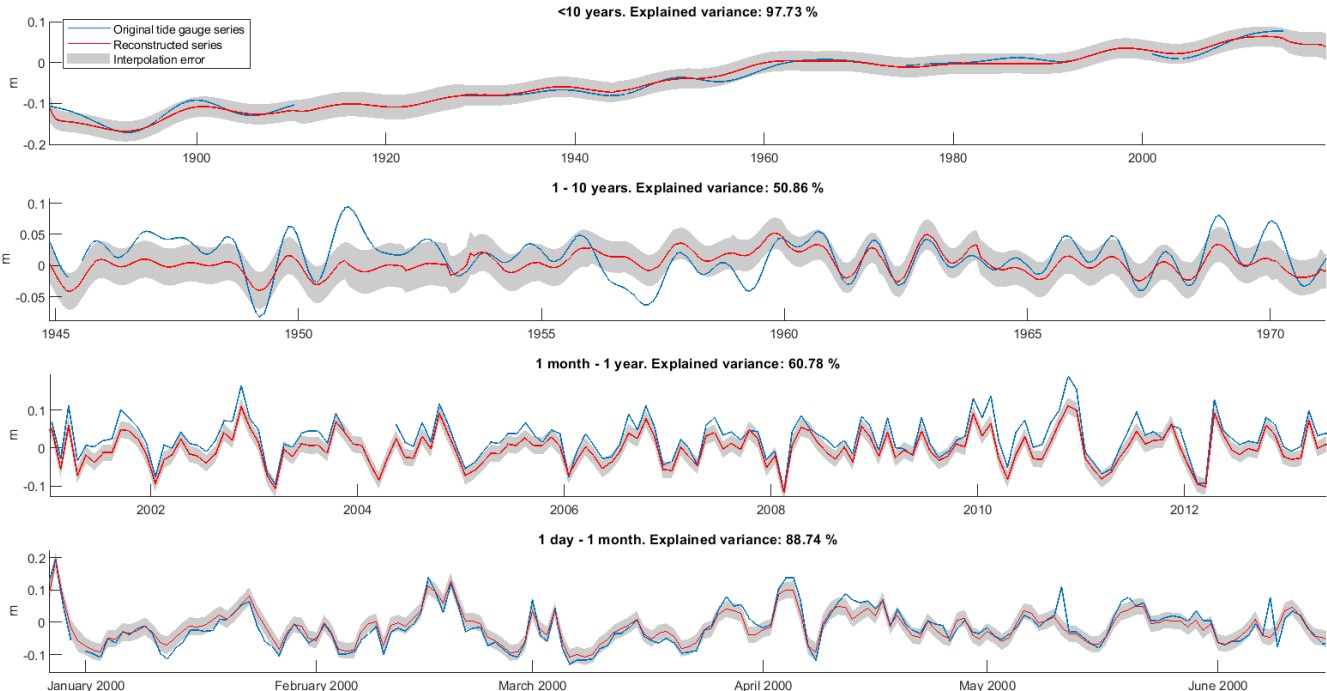

**Figure 6: as for Figure 5, but for the Genoa tide gauge.**

## 4.2 Merging of the reconstructed frequency bands

The cross–validation test has allowed some insight into the capabilities of the method. The next step was to obtain the main result of this work: the reconstructions for each frequency band, now considering all stations (that is, without 295 withdrawing any station, as for the cross–validation test). After that, the reconstructions of the different bands were merged to evaluate the extent to which total sea level can be recovered, and hence to prove that the separation into frequency bands has been carried out correctly.

As an example of the results, Fig. 7 shows, for the interpolation point closest to Barcelona tide gauge: i) the reconstructions in the four frequency bands; ii) a comparison between the original monthly series of Barcelona tide gauge and the merging of the three bands that correspond to periods T>1m; and iii) a comparison between the original daily series 300 of Barcelona tide gauge and the merging of the four frequency bands, which corresponds to periods T>1d. In both cases



there is a high coincidence between the original and the merging of the reconstructed series, showing a correlation of 0.97 for the monthly case, and 0.99 for the daily case.

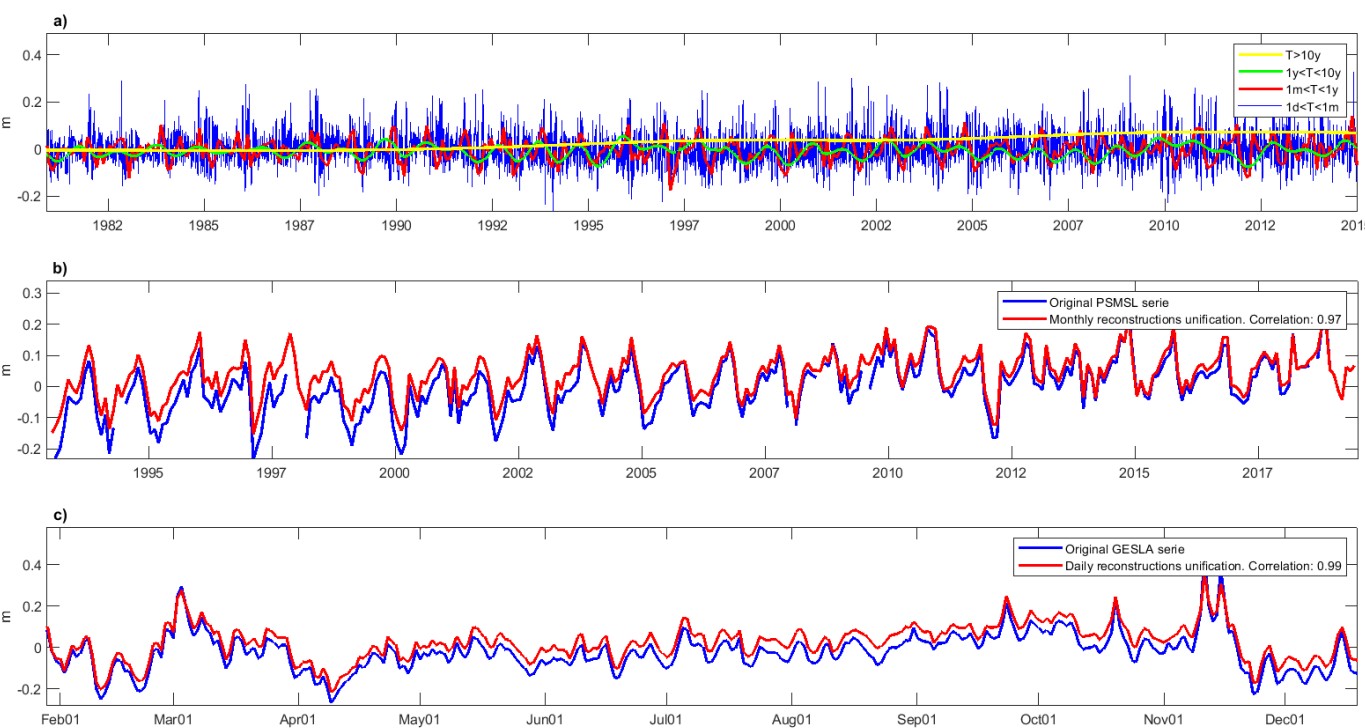

**Figure 7: a) Reconstructions obtained in the four frequency bands for the interpolation point closest to Barcelona tide gauge. b) Monthly merging of the reconstructed frequency bands and the original monthly series of Barcelona tide gauge. c) Daily merging of the reconstructed frequency bands and the original daily series of Barcelona tide gauge. Different time intervals are shown for the monthly and daily series, in order to better appreciate the variability of the merged reconstructions and their differences with the original series.**

Figure 8 shows the interpolation errors of the merged series for the monthly case. Although it has been shown that in some stations actual errors can be higher than the theoretical error estimate, the latter can be useful to reflect the spatial distribution of the interpolation accuracy. The quoted values are actually an average of the interpolation errors over the period from 1884 to 2019, since errors depend on the number of available stations, and this varies with time. The interpolation errors of the daily merged series are also quoted, in this case averaged over the period from 1980 to 2015.

Maximum values of 5.26 cm are obtained for the monthly case, and of 7.05 cm for the daily case. The magnitude of the interpolation errors not only depends on the number of available stations, but also on their location with respect to the considered interpolation point. For this reason, the spatial pattern of the errors clearly shows higher values in regions where no observations are available, such as the North African coast, or where tide gauge series are recent (and hence the average involves time periods when these stations were not available), such as in the Balearic archipelago for the monthly merging.






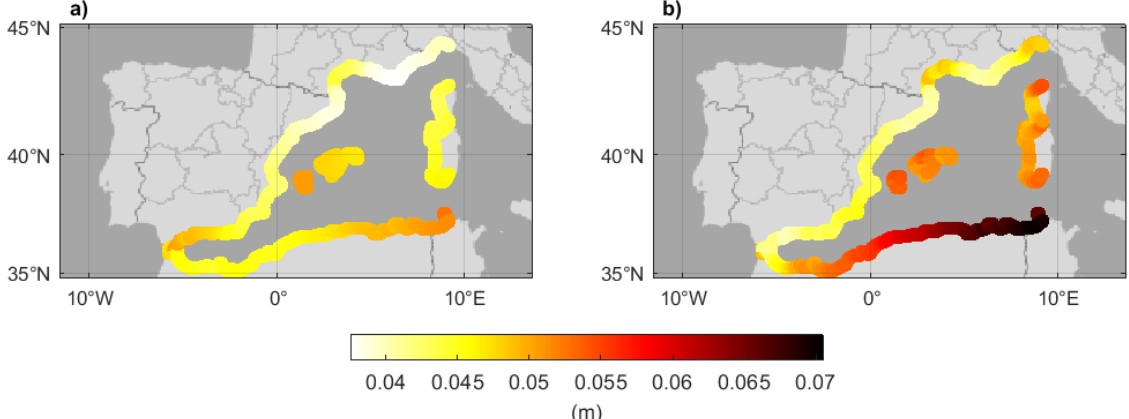

**Figure 8: a) Total interpolation error for the monthly merging of the reconstructions. b) Total interpolation error for the daily merging of the reconstructions.**

## 5 Analysis of the coastal sea level variability in the Western Mediterranean

### 5.1 Reconstructions trends

With the aim of characterising coastal sea level variability, some indicators were estimated from the obtained reconstructions. First, sea level trends were estimated for: i) the merged reconstructed series, for the period covered by altimetry (1993–2019); ii) tide gauge series spanning at least 80% of that period; iii) altimetry series, for the grid points 
closest to the coast. Trends were also calculated for the period (1884–2019). Figure 9 and Table 3 show all these trend values.

For the period covered by altimetry, the trends of the reconstructions and of the altimetry series are similar in magnitude, with a basin–wide mean value of 2.70±0.32 mm/year and 2.45±0.49 mm/year, respectively. However, the values along the 
coast show a smoother continuity for the reconstruction than for altimetry. The heterogeneous coastal trend pattern obtained from altimetry does not seem to make physical sense and could be explained by the limitations of altimetry in coastal areas. When compared with tide gauge trends computed for the period common to the three data sets (Table 3), a good agreement between tide gauges and the reconstruction is obtained except for Algeciras, Barcelona and Tarifa. The trends computed for these stations also show discrepancies with altimetric trends and even with the trends of nearby tide gauges. The lack of 
coherence between tide gauge trends in the Strait of Gibraltar has already been reported by different authors (e.g., Marcos and Tsimplis, 2008; Ross et al., 2000).

The main advantage of the reconstruction over tide gauges is that it allows the estimation of sea level trends along the entire coastline. Moreover, it does not have data gaps, which increase significantly the uncertainty of the trends estimated from tide gauge series. The advantages of the reconstruction over altimetry are that it spans a longer period, provides more 
accurate results and smooths out eventual local anomalous trend values.





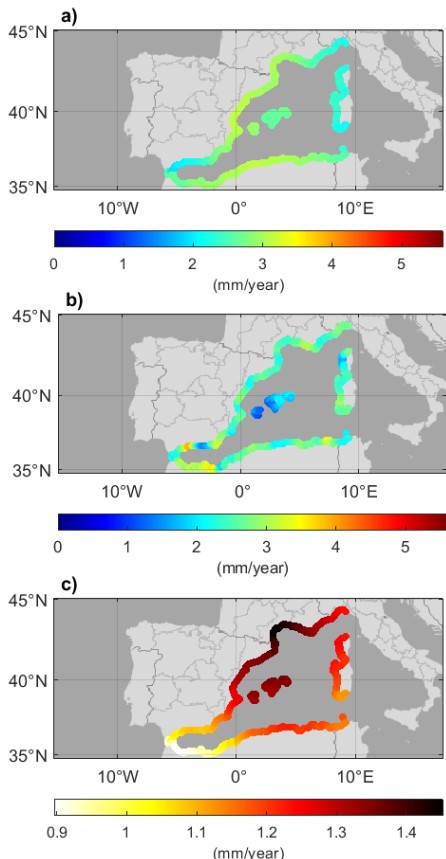

**Figure 9:  a) Sea level trends of the reconstructions for the period covered by altimetry (1993–2019). b) Sea level trends of the altimetry series at the points closest to the coast. c) Sea level trends calculated for the total period of the reconstructions (1884–2019), with a different colour scale.**

**Table 3: Trends of the Tide Gauge, Reconstructions and Altimetry Series, for the Period Common to the Three Data Sets (1993–2019), with the Standard Deviations of the Linear Regression. Only the Stations whose Series are at least 80 % Complete have been included.**

| Station | Tide gauge | Reconstruction | Altimetry |
|---------|------------|----------------|-----------|
| Algeciras | 0.60 ± 0.48 | 2.10 ± 0.42 | 2.21 ± 0.36 |
| Barcelona | 5.59 ± 0.57 | 3.28 ± 0.52 | 2.86 ± 0.48 |
| Ceuta | 1.82 ± 0.41 | 2.51 ± 0.45 | 2.08 ± 0.35 |
| L'Estartit | 2.10 ± 0.51 | 2.97 ± 0.49 | 2.26 ± 0.37 |
| Malaga | 2.17 ± 0.50 | 2.26 ± 0.43 | 3.98 ± 0.39 |
| Nice | 2.85 ± 0.60 | 3.02 ± 0.56 | 2.70 ± 0.46 |
| Sete | 3.53 ± 0.64 | 3.09 ± 0.52 | 2.44 ± 0.42 |
| Tarifa | 4.32 ± 0.42 | 2.35 ± 0.44 | 2.43 ± 0.38 |
| Toulon | 3.05 ± 0.55 | 2.79 ± 0.54 | 3.39 ± 0.38 |
| Valencia | 4.16 ± 0.65 | 3.38 ± 0.54 | 2.37 ± 0.46 |






For the total reconstruction period (1884–2019) the trends range from less than 1 mm/year in the African coasts close to Gibraltar to about 1.5 mm/year in the Gulf of Lions, with a regional mean value of 1.20±0.14 mm/year. This result is consistent with the Mediterranean sea level trend computed from the three stations with the longest series, which is estimated to be between 1.1 and 1.3 mm/year (Gomis et al., 2012), as well as with the global rate of sea level rise for the 20th century,

estimated in between 1 and 2 mm/year (Marcos and Tsimplis, 2007a). The trends show a rapid increase from the 1990s, coinciding with the period covered by altimetry with values ranging from 1.89 to 3.16 mm/yr. For that period the trends estimated from the reconstruction are coherent with those estimated by other authors (e.g. Bonaduce et al., 2016, obtained an average value for the whole Mediterranean basin of 2.44±0.5 mm/y for the period 1993–2012).

**5.2 Sea level variability in different frequency bands**

The seasonal cycle is one of the main components of sea level variability. Seasonal sea level changes are mainly caused by changes in the heat content of the upper layers of the ocean, and by changes in the atmospheric pressure field and winds

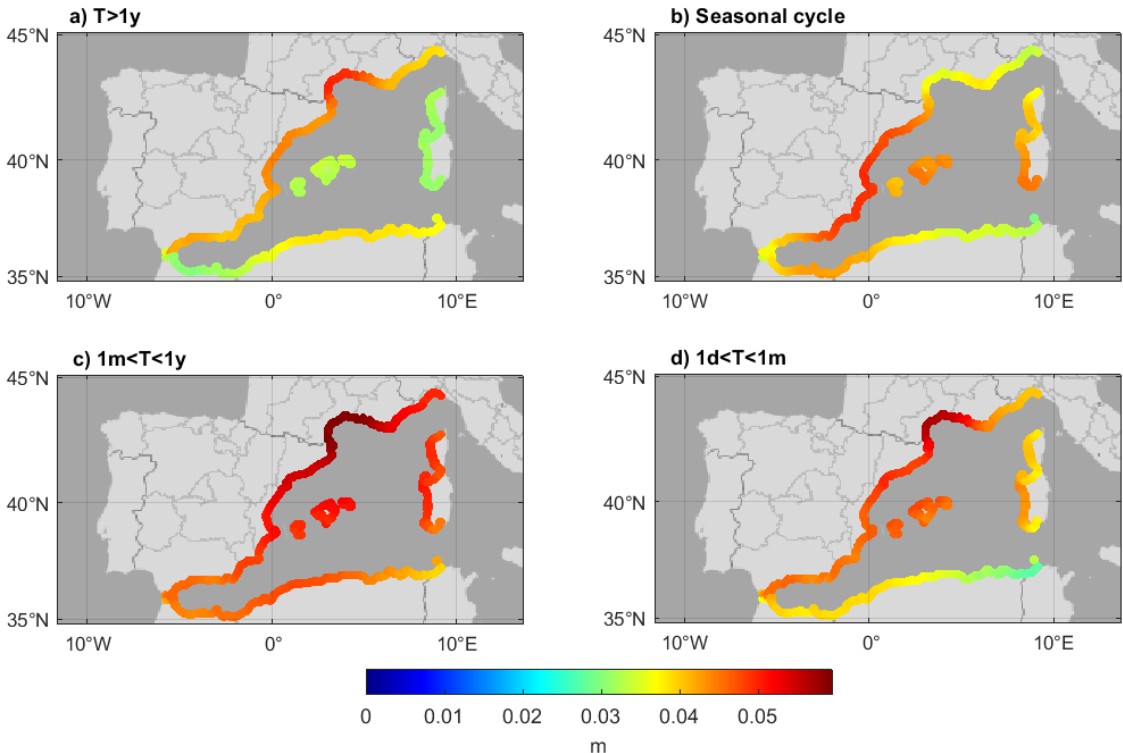

**Figure 10: a) Standard deviation of the reconstructions for the band T>1y, with the seasonal cycle subtracted. b) Standard deviation of the seasonal cycle adjusted from the reconstructions. c) Standard deviation of the reconstructions for the band 1m<T<1y, with the seasonal cycle subtracted. d) Standard deviation of the reconstructions for the band 1d<T<1m, with the seasonal cycle subtracted.**



that modulate the inflow of water from the Atlantic. In the Mediterranean sea, the seasonal cycle is estimated to account, on average, for 20 % of the variance of tide gauge series, and shows significant interannual variability. Also, seasonal cycle variations in coastal areas may differ significantly from those reported in the open ocean (Gomis et al., 2012; Woodworth et al., 2019). The seasonal cycle of the coastal reconstructions, estimated from monthly mean values, accounts on average for 24 % of the coastal sea level variance.

Figure 10 shows the patterns of the variability in different frequency bands quantified in terms of the standard deviation. For the seasonal cycle the variability ranges between 2.92 and 4.97 cm, with a smooth variation along the coast. The largest standard deviations are found at the Eastern coast of the Iberian Peninsula, in agreement with previous authors that located the maximum annual sea level cycle of the western Mediterranean in Alicante (Marcos and Tsimplis, 2007b) and the maximum annual cycle of the atmospheric contribution to sea level in the Alborán Sea (Gomis et al., 2008). The subtraction of the seasonal cycle does not lead to a large reduction of the standard deviation of the reconstruction, which on average is 1 cm lower without the seasonal cycle. The deseasoned reconstruction has an average standard deviation of 3.77 cm for periods T>1y, 4.91 cm for periods 1m<T<1y and 4.27 cm for periods 1d<T<1m. In all frequency bands, the highest standard deviations of the deseasoned series are obtained in the Gulf of Lion.

## 5.3 Influence of atmospheric climate modes on sea level variability

The four main atmospheric modes driving western Mediterranean sea level variability are NAO, EA, EA/WR and SCAN. Their influence has already been studied among others by Martínez-Asensio et al., 2014, who used long tide gauge series from the whole basin, as well as altimetry products. Our coastal reconstructions allow us to complement the study of the influence of these climate modes in two ways: first, by covering the entire coastal region (i.e., not only where tide gauge records are available), and second, avoiding the use of coastal altimetry. In addition, the sea level series of our reconstructions cover a longer period than altimetric products, thus enabling the analysis to extend to the whole period covered by climate indices (since 1950).

Figure 11 shows the correlation patterns between the monthly coastal reconstruction and the four climate indices. Correlations have been calculated both for the complete series, and for the seasonal mean values of the reconstruction and the indices, with winter accounting for January–March, spring for April–June, summer for July–September and autumn for October–December.

For the total series, the NAO index is anti–correlated with sea level since the western Mediterranean participates of the subtropical high pressures and hence sea level lowers when the NAO index is in a positive phase. Conversely, the EA and SCAN indices show a positive correlation. In winter, two indices dominate sea level variability: the NAO index, with correlation values below -0.6 at some points and showing significant correlations along the entire basin coastline, and the SCAN index, with positive correlations (the NAO and SCAN indices are interdependent and anti–correlated with each other). The obtained correlation patterns are similar to those obtained from open–ocean altimetry for the period 1993–2010 (Martínez-Asensio et al., 2014), although the values obtained for the coastal reconstructions are somewhat weaker for the





two dominant indices. In spring, the EA index clearly dominates over the others, being the only one with significant
correlations throughout the basin, with positive correlation values above 0.3.

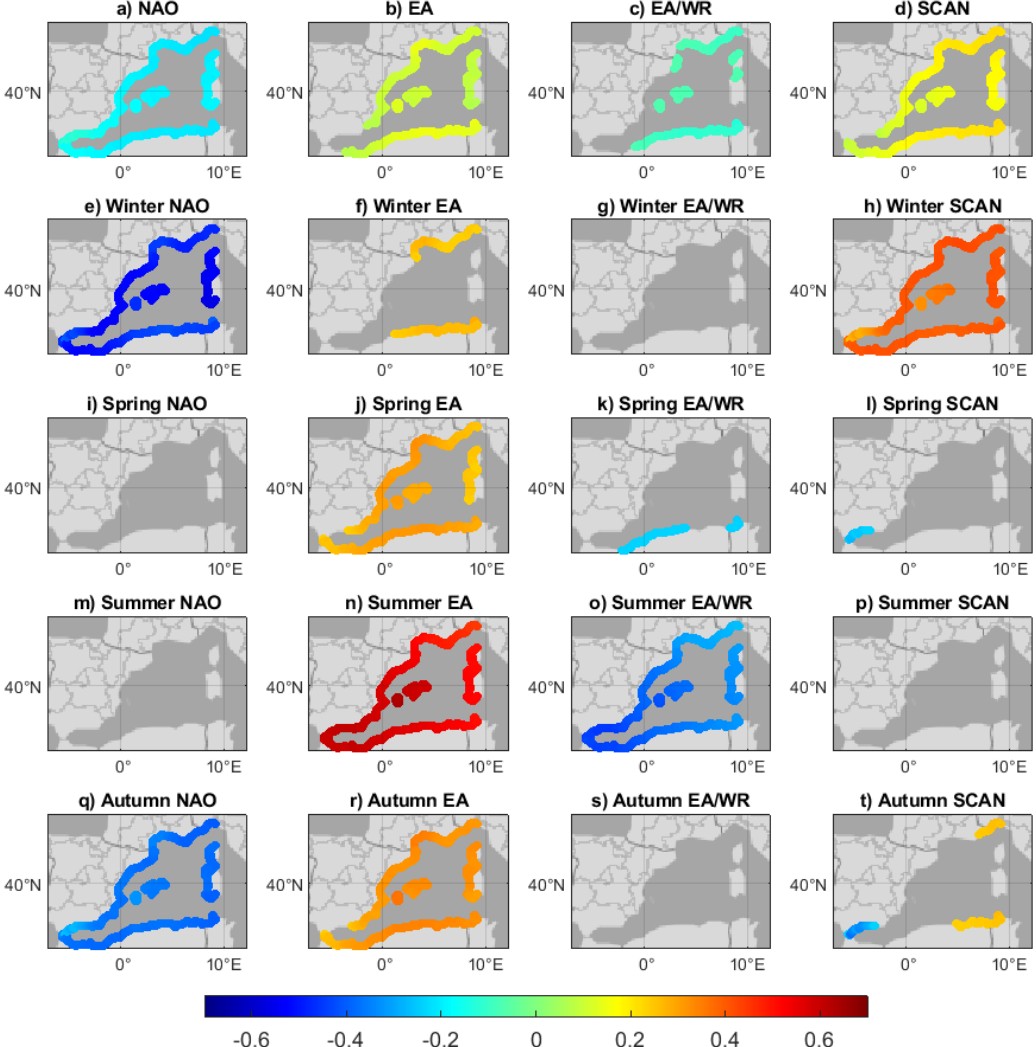

**Figure 11: Seasonal and total maps of correlation coefficients between climate indices and the coastal sea level reconstruction for the period 1950–2019. Only correlations significant at the 95 % level have been plotted.**

In summer, the correlation patterns obtained for our reconstructions clearly differ from those obtained basin–wide for the
barotropic hindcast forced by atmospheric pressure and wind used by Martínez-Asensio et al. (2014). During that season the
EA is the dominating index, with positive correlations up to 0.5 basin–wide, while the EA/WR index shows negative
correlations of around -0.3. These results contrast with the negative correlations for the EA index and the positive
correlations for EA/WR obtained Martínez-Asensio et al. (2014). The reason could be that in the western Mediterranean the



EA pattern is mostly related to freshwater fluxes, while EA/WR is mostly related to heat fluxes and has also an impact on precipitation. That is, both indices mostly influence the steric and mass components of sea level, while the barotropic

hindcast used by Martínez-Asensio et al. (2014) only accounts for the forcing of atmospheric pressure and wind. Finally, in autumn it is the NAO index that seems to dominate, with correlations around -0.5, followed by EA, with values up to 0.4.

## 6 Discussion

A major objective of this work was to explore whether using tide gauge data in an optimal way could result in a coastal sea level dataset more accurate than current coastal altimetry products. Figure 12 shows, for the different frequency bands,

the correlations between the daily reconstructions resulting from the cross–validation test (i.e., withdrawing from the input observations the tide gauge record that is intended to be reproduced) and the original tide gauge series, as well as the correlations between altimetry (at the closest grid point to the tide gauge) with the DAC applied, and the original tide gauge series. Correlations have been computed for the period covered by both altimetry and our reconstructions, that is, from 1993 to 2015.

Figure 12 shows that the correlations are in general significantly higher for the reconstructed series than for the corrected altimetry for all frequency bands. It should be kept in mind that these correlations have been calculated for the period (the last decades) when a larger number of observations in all bands are available (this also explains why the correlations shown in Fig.12 are higher than those shown in Fig. 4 for the whole period of the reconstruction). Namely, the correlations between the reconstructions obtained through the cross–validation test and the tide gauge series are higher than 0.5 at all stations and

frequencies, being higher than 0.75 in most of the stations. Conversely, for altimetry with the DAC applied and for the frequency band for which it performs better (1d<T<1m), correlations are all lower than 0.5 (for the other frequency bands correlations are much lower). It is worth mentioning that in this band the results are better because the variability in that frequency is dominated by the atmospheric mechanical forcing, which is reasonably well modelled by DAC. More precisely, the average correlations between coastal reconstructions and tide gauges are 0.95 for the T>10y band, 0.83 for the 1y<T<10y

band, 0.92 for the 1m<T<1y band, and 0.91 for the 1d<T<1m band. On the other hand, the average correlations between the altimetry series with the applied DAC and the tide gauge series are -0.25 for the T>10y band, 0.08 for the 1y<T<10y band, 0.02 for the 1m<T<1y band, and 0.43 for the 1d<T<1m band. This confirms that using tide gauge data in an optimal way allows the retrieval of coastal sea level with a significantly higher accuracy than using altimetric products for all time scales. The proposed method to estimate coastal sea level can be applied in a straightforward way to any other region, keeping in

mind two potential limitations. The first one is that the correlation elements of the Optimal Interpolation matrices should be reliable, and this implies the existence of a reliable, long enough sea level data with high spatio–temporal resolution, such as the outputs of the SOCIB model in our case. Otherwise, the correlation matrices will have to be calculated through the fitting of analytical functions, which is usually less accurate. The second limitation is that the quality of the reconstruction will also depend on the spatial distribution of tide gauge observations. A relevant advantage of the method is that given the spatial

distribution of tide gauges, the interpolation errors can be estimated a priori. Although the theoretical error estimate may be



optimistic (due to the assumption that the correlation matrix elements are fully representative of actual correlations), it usually provides a reliable error pattern. Moreover, having in mind that the theoretical interpolation error constitutes a lower boundary for actual errors can be useful to decide about the application or not of the method.

Regarding previous efforts to retrieve sea level in the region, all previous reconstructions gave greater emphasis to the open ocean and have the limitation of relying on altimetric products when coastal sea level is attempted to be reproduced. To our knowledge, this is the first attempt to obtain a reconstruction specific to the coastal region.

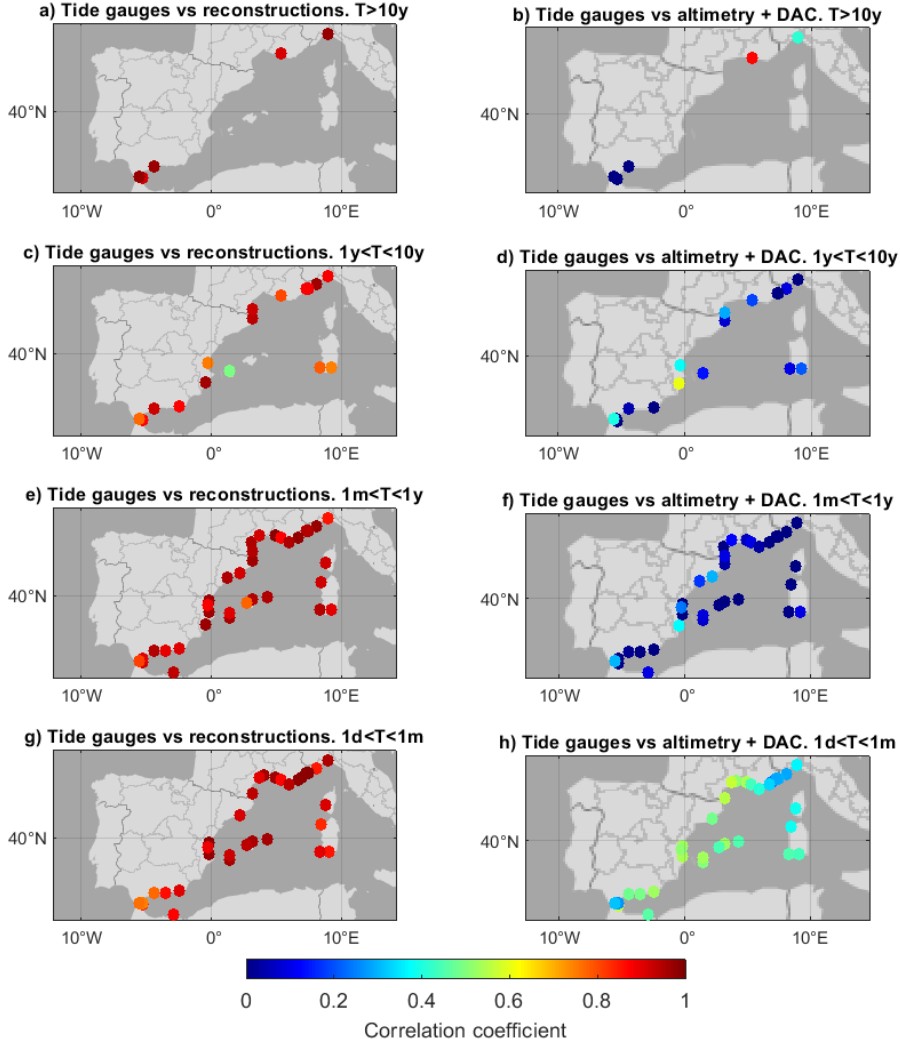

**Figure 12: Correlations between the reconstructions obtained through the cross–validation test and the original tide gauge series (left), and correlations between the atmospherically corrected altimetry series and the original tide gauge series (right), for the four frequency bands.**



## 7 Conclusions

Sea level reconstructions have been obtained for the whole coast of the western Mediterranean basin by applying an Optimal Interpolation scheme to tide gauge observations. The reconstructions have been obtained for four frequency bands, and then merged to obtain total sea level. In order to validate the robustness of the method, a cross–validation test was applied using the tide gauge series themselves as independent observations. The test was applied to each frequency band, giving successful results except at a few particular stations (e.g., near the Strait of Gibraltar). It was also checked that the merging of the reconstructions obtained in the four frequency bands accurately recovers the original total sea level series at coastal points close to tide gauges.

A major conclusion of the work is that the reconstructions provide significantly better estimates of coastal sea level than current altimetry products with the atmospheric correction added back. This has been proved again via cross–validation, by obtaining the reconstruction nearby each tide gauge location with a prior withdrawal of that tide gauge record from the interpolation scheme.

The reconstructions have been used to gain some insight in different aspects of coastal sea level variability. Thus, coastal trend values have been calculated for the period (1884–2019). Also, trends computed for the period covered by altimetry are fairly consistent with those obtained from altimetry data, but the pattern of the trends along the coast shows a smoother continuity for the reconstructions. It has also been found that the relationship of sea level and climate indices obtained by Martínez-Asensio et al. (2014) are generally comparable with those obtained from our reconstructions, but they show noticeable discrepancies in summer (the signs of the correlations with the EA and EA/WR indices are inverted) likely due to the type of sea level product used by Martínez-Asensio et al. (2014).

In summary, results indicate that it is possible to obtain accurate coast–wide sea level series from an optimal processing of tide gauge observations only. The accuracy of the reconstruction has been shown to be vary regionally, depending on the number of available stations and also likely on the more or less correct representation of the correlation elements of the Optimal Interpolation matrices. The applicability of the method to other regions is conditioned, first, by the availability of long enough sea level datasets with the required spatiotemporal resolution to compute reliable correlation functions; the number of available tide gauge observations and their spatial distribution will also affect the accuracy of the reconstruction.

## 8 Appendices

### Appendix A: Validation of the correlations inferred from SOCIB model outputs for the optimal interpolation of coastal sea level

In order to validate the use of the correlations of the numerical model outputs in the implementation of the optimal interpolation, gaussian noise was added to the model series, whose variance was adjusted to carry out a first reconstruction test using these series as pseudo–observations, to verify the ability of the reconstruction method. The differences between the



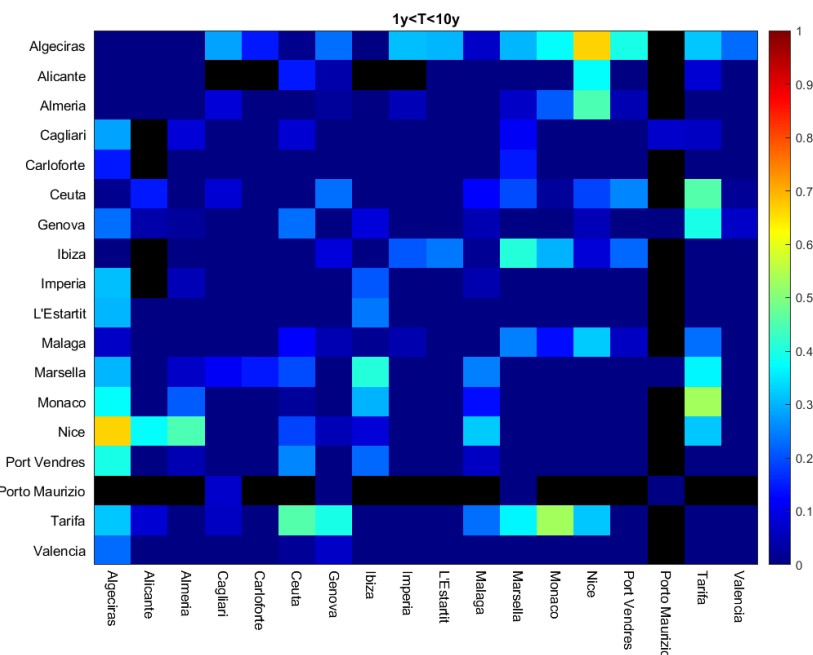

**Figure B1: Differences between the correlation between tide gauge series and the correlation between SOCIB model series (with random noise added) at the points closest to each tide gauge, for the frequency band 1y < T < 10y.**

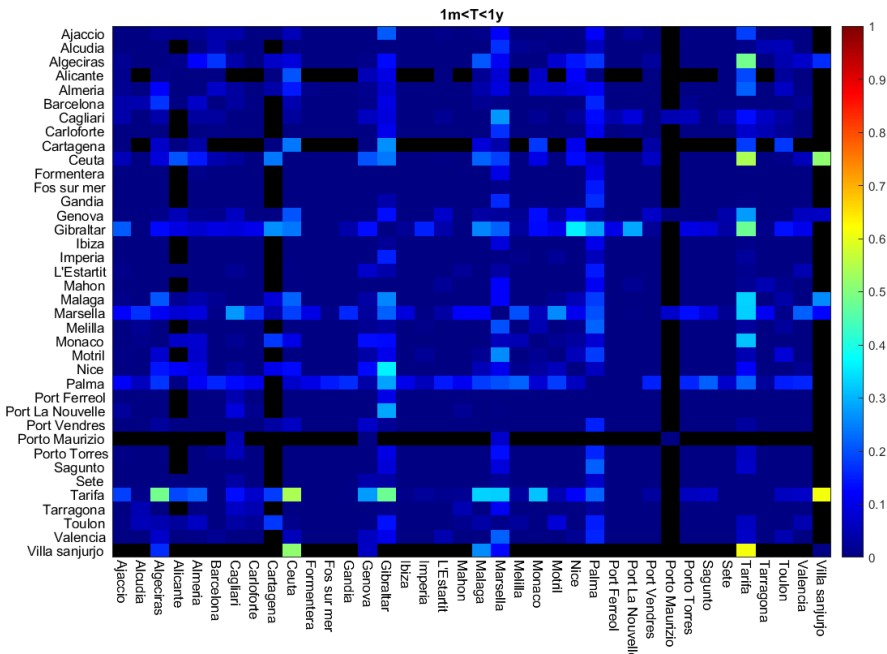

**Figure B2: Differences between the correlation between tide gauge series and the correlation between SOCIB model series (with random noise added) at the points closest to each tide gauge, for the frequency band 1m < T < 1y.**





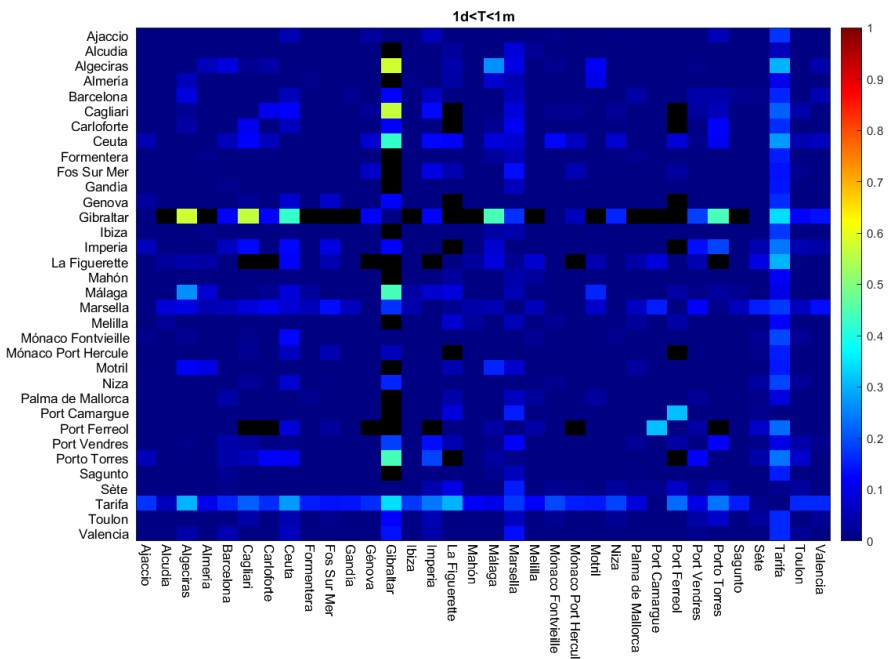

**Figure B3: Differences between the correlation between tide gauge series and the correlation between SOCIB model series (with random noise added) at the points closest to each tide gauge, for the frequency band 1d < T < 1m.**


correlation patterns between the tide gauge series and correlation patterns between the model series with noise (for the points closest to the tide gauges) have been included in this document.

Figures B1, B2 and B3 show the differences between the correlations between pairs of tide gauges and the correlations between pairs of SOCIB model series located at the closest point to each tide gauge. Gaussian noise has been added to the

model series, with an error variance being optimised to minimise the differences with respect to tide gauge correlations for each frequency band. Correlations that could not be calculated due to the shortness of the time period spanned by the two series are shown in black.

**Appendix B: Statistical interpolation errors associated to the reconstruction of each frequency band**

Statistical interpolation errors associated to the reconstruction of the four frequency bands are shown in Figure B4. The displayed values are the average of the errors along the period spanned by the reconstruction, since errors vary with time due to the variation of the number of tide gauge series available. The spatial distributions of the errors indicate that these are larger in areas where no observations are available, or where tide gauge series are shorter.




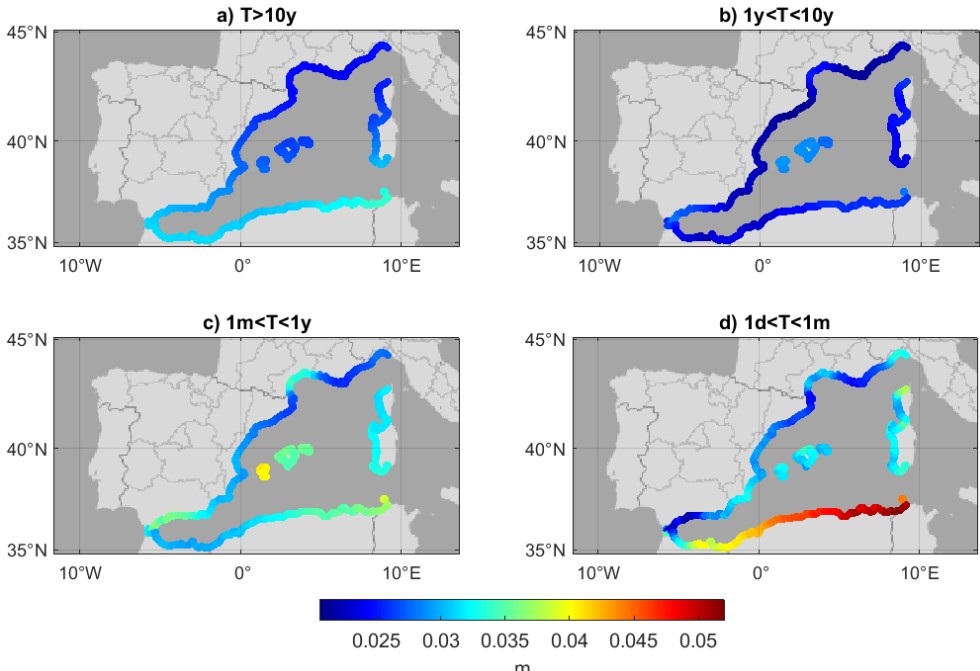

**Figure B4. Temporal average of the analysis error (in m) (a) for the band T>10y (b) for the band 1y<T<10y, (c) for the band 1m<T<1y and (d) for the band 1d<T<1m.**


## 9 Data availability

The coastal sea level reconstructions data for the western Mediterranean Basin developed in this work will be available at PANGAEA Data Publisher. The DOI will be provided later. Tide gauge data are available from Global Extreme Sea Level Analysis project (http://www.gesla.org/; Caldwell et al., 2015; Haigh et al., 2021; Woodworth et al., 2016), and from the

Permanent Service for Mean Sea Level (PSMSL; https://www.psmsl.org/). WMOP numerical model outputs are available through the Balearic Islands Coastal Observing and Forecasting System data center (SOCIB; https://www.socib.es/?seccion=dataCenter). The satellite altimetry data are available through the Copernicus Marine Environment Monitoring Service (CMEMS, https://resources.marine.copernicus.eu/?option=com_csw&view=details&product_id=-

SEALEVEL_MED_PHY_L4_REP_OBSERVATIONS_008_051), and the Dynamic Atmospheric Correction data data are available through the Archiving, Validation and Interpretation of Satellite Oceanographic (AVISO; https://www.aviso.altimetry.fr/en/data/products/auxiliary-products/dynamic-atmospheric-correction/description-atmospheric-corrections.html). The Climatic indices data are available through the NOAA Climate Prediction Centre website (http://www.cpc.ncep.noaa.gov/data/teledoc/telecontents.shtml).





## 10 Author Contributions

DG and GJ conceived the study. JR analysed the entire dataset, did the computational work, and wrote the first draft of the manuscript. DG and GJ developed most of the methodology and devised the structure of the article. All authors contributed to the interpretation of the results, and to the final version of the manuscript.

## Acknowledgments

This work is part of the R+D+I project VENOM (PGC2018-099285-B-C21, PGC2018-099285-B-C22), funded by MCIN/AEI/10.13039/501100011033 and by ERDF A way of making Europe. We thank J. Villalonga and M. Agulles for their comments and computational help. The authors confirm that there are no known conflicts of interest associated with this publication.

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
