# Peer review of "Reconstruction of Mediterranean coastal sea level at different timescales based on tide gauge records"

_EGUsphere, 2022_

## Author Response (AR1)

**Point-by-point reply to Referee #1**

**C1:** Review of "Reconstruction of Mediterranean coastal sea level at different time scales based on tide gauge records" by Alcantara et al., under discussion in Ocean Science.

In this manuscript the authors combine sea level observations from tide gauges and satellite altimetry with output of an ocean model in order to produce a costal sea level reconstruction with high spatial and temporal resolution along the western Mediterranean coast.

The paper will reads very well, the methodology is clearly explained and the results quite relevant for a broad range of scientists and policymakers. In particular, the analysis of sea level variability over four different frequency bands is rather interesting and useful to understand the physical processes behind the observed changes.

I recommend the manuscript for publication in Ocean Science after a few minor issues have been addressed. My comments follow their chronological order and they are not sorted by relevance.

**R1:** We would like to thank the referee for the effort of revising our work and we are grateful for the appreciation of the usefulness and interest of our results. We have taken careful note of his/her comments, which are answered below; they have undoubtedly helped to improve the new version of the manuscript.

**C2:** Line 44: "without some further data processing" is a quite vague statement that could be followed by a short explanation of what this data processing generally includes.

**R2:** The reviewer is right. A brief explanation of the data processing that is commonly applied to study regional coastal sea level forcings has been included in the new version of the manuscript. We have rewritten the sentence and now it reads (lines 44 – 47): *"These have a spatio–temporal variability that is not always captured by the current observational networks and some additional information is required (i.e., running ocean barotropic models forced with the available atmospheric pressure and wind reanalyses, in order to resolve the small scales not captured by the sea-level network; Carrère and Lyard, 2003)."*

**C3:** Line 75: please add a reference for the optimal interpolation method.

**R3:** We have added two references, one for a pioneering application of OI to oceanographic data and another one more focused on the method itself:
- *Bretherton, F. P., Davis, R. E., and Fandry, C. B.: A technique for objective analysis and design of oceanographic experiments applied to MODE-73\*, Deep Sea Research and Oceanographic Abstracts, 23, 559–582, https://doi.org/https://doi.org/10.1016/0011-7471(76)90001-2, 1976.*
- *Pedder, M. A.: Interpolation and Filtering of Spacial Observations Using Successive Corrections and Gaussian Filters, Mon Weather Rev, 121, 2889–2902, https://doi.org/https://doi.org/10.1175/1520-0493(1993)121<2889:IAFOSO>2.0.CO;2, 1993.*

**C4:** Table 1: it would be nice to show the location of all stations in a figure; alternatively, the authors could at least add a few labels to Figure 1, with the names of those stations that are explicitly discussed later in paper.

**R4:** We have updated Figure 1 by adding labels with the names of the stations discussed throughout the manuscript.

**C5:** Line 98: please add a few more details about how datum shifts are correct for, the current sentence is quite concise.

**R5:** In fact, this correction was only applied to the Cagliari tide gauge of GESLA-2; below we show a plot of this series before and after the applied correction. PSMSL series belong to the group of "Revised Local Reference" tide gauges and therefore they are not expected to have datum changes. Conversely, the tide gauges of GESLA-2 do not provide specific documentation on their vertical local reference, and although they have been submitted to some quality control, many have obvious datum changes (this is often the case for eastern Mediterranean tide gauges, for instance).

For Cagliari we identified two different datums, before and after 28 January 1993 (marked with a red line in the plots). Each of the two periods is subtracted from its mean value, thus converting them into sea level anomalies. We are aware that this procedure is rather simple, but it is effective in most cases, as shown by other authors (see e.g., Church et al 2004: https://doi.org/10.1175/1520-0442(2004)017<2609:EOTRDO>2.0.CO;2) and preferable to discard part of the series.

In the new version of the manuscript, lines 100- 102 now it reads: **"*Only Cagliari tide gauge series showed an obvious datum shift; this made necessary to visually identify the intervals with different datums and subtract their means separately, in order to convert the original data into zero-mean anomalies.*"**

[Figure]

**C6:** Line 185: change "series" into time series or stations.

**R6:** The word "series" has been replaced by "stations" in the new version of the manuscript.

**C7:** Line 189: how is it possible to use the frequency bands from the previous point when different stations are used? Are the frequency bands determined on the ensemble of stations, hence valid for the whole domain? This issue could be made more explicit.

**R7:** For each frequency band, the reconstruction is obtained over the entire reconstruction grid (the coastal points of the SOCIB model). The 'previous' frequency band subtracted from

each tide gauge in order to reconstruct the next is that obtained from the lower frequency reconstruction at the nearest grid point. The separations between tide gauges and their nearest grid point are rather small; for example, for the tide gauges used to obtain the frequency band 1m<T<1y, the average separation is 1.4 km, and the maximum is 4.1 km. It must also be clarified that prior to each subtraction, both the original series and the reconstructed series from the previous band are converted into anomalies with respect to their mean for the common period. This detail is important in order to keep consistency in the long term evolution.

In the revised version of the manuscript, we have added (lines 192 - 201):

- *"For the band 1y<T<10y, tide gauges with at least 10 years of consecutive data were considered (see Fig. 1). First, the reconstruction obtained in the previous step in the nearest grid point to the tide gauge was subtracted from the original series and then the frequencies corresponding to periods T<1y were removed, also by means of a Butterworth filter of order 10.*
- *For the band 1m<T<1y, all available PSMSL tide gauges were considered (see Fig. 1), and the two frequency bands reconstructed in the previous steps were removed from the original series. As these consisted of monthly data, there was no need to remove the periods T<1m.*
- *For the 1d<T<1m band, the three previous reconstructions (obtained from PSMSL data) were subtracted from each of the GESLA–2 series (this required a prior conversion of the three bands to daily values by means of linear interpolation)."*

**C8:** Line 217: please add reference for equation 5.

**R8**: The reference for this equation has been included three lines below (lines 228 - 230) and now it reads as **"*The second part of the expression corresponds to a temporal correlation between observations (it is based on the exponential functions typically used to define the weights of correlation matrices, see e.g., Pozzi et al 2012) and is not always used in Optimal Interpolation"*.** By the way, we have taken advantage to correct two typos in this formula: the spatial distance between stations $d_{ij}$ must be squared and the temporal distance must be always a positive value $/dt/$. Equation 5 now reads:

$$\theta_{ij} = e^{-\frac{d_{ij}^2}{2L_s^2}} \cdot e^{-\frac{|dt|}{L_t}}$$

**C9:** Line 223: I understand the need to combine observations from different times, but it rests on the assumption that sea level variability is constant over time, which might not be the case. This issue warrants a more explicit discussion.

**R9:** We appreciate the warning, however we do not strictly assume that sea level is constant over time. First it must be noted that this adjustment is only applied for the lower frequency band (T>10 years), and the assumption is that at these timescales sea level variability is slow enough to consider that what happened within the two years forward and backward provides relevant information on what is happening at the current time. Please note moreover that backward and forward observations are weighted: their influence on the reconstruction decreases with their temporal separation from the instant under consideration.

**C10:** Line 272: since some stations are not reproduced very well by the reconstruction, it might be worth removing them from the final product. I wonder whether the authors have tried this. If not, they might want to discuss why they choose to keep all stations.

**R10:** In absence of clear errors in the series, the disagreement between some stations and the reconstruction must be attributed to the presence of local forcings which are not shared with the neighbouring stations. This is probably the case of the stations near the Strait of Gibraltar, for instance. Discarding 'anomalous' stations from the reconstruction would hide this feature, as the dominant forcings would be projected also at the location of these stations. We sincerely think that it is more honest keeping them and accepting that the reconstruction cannot cope with all the variability, particularly if it is of very local scale.

**C11:** Line 305: please explain what do you mean by interpolation errors.

**R11:** The optimal interpolation method not only provides interpolated values, but also a statistical estimate of the errors involved in the process. These errors (due to errors in the observations and to the discrete sampling of the field) are quantified by formula (4) of section 3.1. Note that the reference *Gomis and Pedder (2005)* has been included before equation (4), in line 173. More specifically, the referred errors in this line are those calculated at each point and for each band, averaged over the period 1884-2019 for the monthly case and for the period 1980-2015 for the daily case, and unified as explained in in the second paragraph of Section 3.3.

In the new version of the manuscript, lines 312 - 315 now read: "*Figure 8 shows the average interpolation errors of the merged series for the monthly case. Although it has been shown that in some stations actual errors can be higher than the theoretical error estimate, the latter can be useful to reflect the spatial distribution of the interpolation accuracy. The quoted values are an average of the interpolation errors over the period from 1884 to 2019, since errors depend on the number of available stations, and this varies with time.*"

And the title of Fig. 8 will read: "*Figure 8: a) Average interpolation error for the monthly merging of the reconstructions. b) Average interpolation error for the daily merging of the reconstructions.*"

**C12:** Line 332: This advantage of the reconstruction could be highlighted better in the abstract.

**R12:** In the new version of the manuscript, lines 18 - 21, which belong to the abstract, now read: "*The obtained reconstructions allow to characterize the coastal sea level variability, to estimate coastal sea level trends along the entire coastline and to examine the correlation between Western Mediterranean coastal sea level and the main North Atlantic climate indices.*"

**C13:** Line 345: most recent global reconstructions since Hay et al. (2015), especially those by Dangendorf and colleagues, actually estimate trends smaller than 1.5 mm/yr (more precisely, about 1.2 mm/yr until 1990 and about 1.6 mm/yr until the mid 2010s). The reference to Marcos and Tsimplis (2007a) is outdated.

**R13:** In the new version of the manuscript, lines 348-349 read: "*…as well as with the global rate of sea level rise for the 20th century, estimated through various reconstructions between 1.3 and 2 mm/year (Dangendorf et al, 2017).*"

**C14:** Figure 12: I wonder whether the fact that the reconstruction is better correlated to tide gauges than satellite altimetry is not simply a direct consequence of the applied methodology. The author should be cautious in arguing that such a correlation is a proof that their construction is superior to satellite altimetry. I'm not saying that I disagree, but such a claim requires a more detailed discussion.

**R14:** While tide gauge observations are obviously not error-free, they are currently the most reliable source of sea level data. For this reason, it seems reasonable that the reliability of coastal sea level products is cross-checked against tide gauge series. On the other hand, tide gauges are limited by being point-wise measurements; filling this gap is precisely a major objective of our work.

We honestly believe that the fact that tide gauges are better correlated with our reconstruction than with satellite altimetry, is not a direct consequence of the applied methodology. We recall how cross-validation tests work: the reconstructions used to correlate with a given tide gauge are obtained from all available observations **except** the tide gauge under consideration (i.e., we assume a data void at the tide gauge location). If this correlation is significantly higher for all frequency bands than the correlation between the tide gauge and altimetry, it seems reasonable to assume that this can also be the case where no observations are available.

The advantage of the reconstruction over altimetry is probably not only a consequence of the goodness of the method, but also of the significant limitations of altimetry at the coast.

**Point-by-point reply to Referee #2**

**C1:** This study performed the reconstruction of coastal sea level variability of the Mediterranean Sea from tide-gauge datasets using an optimal interpolation method. The authors showed that the reconstruction provides better estimate of coastal sea level variability than the altimeter data. The topic of the study is important, and the method is well thought out. Hence, this manuscript is acceptable after revisions.

**R1:** We thank the referee for his/her comments as well as for the effort in revising our work. All his/her comments have been taken into account and addressed in the present document. An improved version of the manuscript has been resubmitted, thanks to this discussion.

**Major comment:**

**C2:** L327-328: As the authors mentioned, there are large differences of the sea level trends between the tide-gauge data and the reconstruction at Algeciras, Barcelona and Tarifa (Table 3). What is the reason of these differences? In addition, the trend of the tide-gauges is more heterogeneous than that of the reconstruction (Table 3). The optimal interpolation method for the reconstruction might not be suitable to capture small-scale coastal processes. Please discuss this point.

**R2:** The lack of coherence between the trends calculated through the historical series of tide gauges located in the region close to the Strait of Gibraltar has been reported by several authors (e.g., Ross et al., 2000; Marcos and Tsimplis, 2008). This lack of coherence is maintained during the period covered by altimetry, and important discrepancies have also been reported between the trends provided by altimetry and those shown by some tide gauges, such as Tarifa or Barcelona (Taibi & Haddad 2019: https://doi.org/10.1007/s00343-019-8164-3). In absence of clear errors in the series, we

agree with the reviewer in attributing the observed lack of coherence to the presence of local forcings which are not shared with the neighbouring stations.

The statistical interpolation applied in our work relies on the existence of meaningful spatial correlations, obtained in our case from a numerical model. If a given station is submitted to local forcings that are not well captured by the model, the elements of the correlation matrices affecting that station will not be accurate and the interpolation will fail in the vicinity of that station, as stated in in the last paragraph of Section 4.1: *"For these frequency bands, the differences between the original and the reconstructed series are larger than the statistical interpolation error given by the Optimal Interpolation formulation (Eq. (4)); this suggests that for some stations the correlation elements of the Optimal Interpolation matrices are not correctly represented"*.

In summary, we agree with the reviewer in that the optimal interpolation method used for the reconstruction might not be suitable to capture small-scale coastal processes. Consequently, we have highlighted this in the new version of the conclusions (lines 461 - 466): **"The accuracy of the reconstruction has been shown to vary regionally. The level of accuracy depends on the number of available stations and also on the accuracy of the representation of the correlation elements of the Optimal Interpolation matrices which in our case are provided by a numerical model. The applicability and performance of the method to other regions is conditioned, first, by the availability of long enough sea level datasets with the required spatio-temporal resolution to compute reliable correlation functions, and second, to the number of available tide gauge observations and their spatial distribution"**.

**Minor comments:**

**C3:** Section 2.1: How do you remove the astronomical tide from the tide-gauge data?

**R3:** We do not remove the astronomical tide from the tide-gauge data, but by using daily averages we assume that the tidal signal has mostly been filtered out.

**C4:** L264: "interdecadal"?

**R4:** In the new version of the manuscript, line 271 now read: **"For the interannual to decadal frequency band (1y<T<10y) the reconstructions explain..."**

**C5:** L388-389: In this paragraph, the authors compared your result in summer with the result of the barotropic model by Martinez-Asensio et al. (2014). This comparison does not make sense. The authors should compare your result with the result of the tide-gauge data by Martinez-Asensio et al. (2014). The authors have to mention the difference of the results and advantages of your reconstruction.

**R5:** The reviewer is right, thanks for the suggestion. In the new version of the manuscript, the last paragraph of section 5.3. reads: **"In summer, the correlation patterns obtained from our reconstructions differ slightly from those obtained from tide gauge series in the western Mediterranean by Martínez-Asensio et al. (2014). Our results show that, during summer, the EA is the dominating index, with positive correlations up to 0.5 basin-wide, while the EA/WR index shows negative correlations of around -0.3. Martínez-Asensio et al. (2014) also obtain positive correlations between the tide gauges and the EA index, but always lower than 0.5, and often not significant; for the EA/WR index they obtain non-significant negative correlations. Overall, the sea level reconstruction suggests a greater influence of the EA and EA/WR indices (mainly related to freshwater and heat fluxes) on western Mediterranean sea level variability in summer than what is obtained from**

*pointwise observations. Finally, in autumn it is the NAO index that seems to dominate the variability, with correlations around -0.5, followed by EA, with values up to 0.4.*"

**C6:** Legend of Fig. 5: "serie" => "series"

**R6:** The legend of Figure 5 has been corrected, thanks.

**List of relevant changes in the manuscript**

- New affiliation and new project: recently, contact author Jorge Ramos, has started a new contract at the Instituto Oceanográfico de Baleares. The new affiliation has been added (line 3), as well as the mention to the new project in the Acknowledgements (line 511), and in the Financial Support (line 516).

- The ability of the reconstructions to provide a dataset covering the entire coastline has been highlighted in the abstract (lines 18 - 21).

- In the introduction, a brief explanation of the data processing generally applied to coastal sea level forcings, whose spatio-temporal variability is not always well captured by the current observational networks, has been included (lines 44-47)

- Two references have been added after the first mention in the manuscript to the Optimal Interpolation method (line 82).

- It has been clarified that the datum shift corrections were necessary exclusively for the Cagliari tide gauge, and the explanation of how this was done has been improved (lines 100 - 102).

- Regarding the accessibility of the altimetric products (in the Data sets section in lines 137 - 139, and in Data availability, lines 496 - 498), the link that was no longer operational has been discarded, and instead the product identifier where the data used were included is specified, as well as the new product ID where they are currently included.

- We have added to equation (4), which quantifies the interpolation errors, the reference *Gomis and Pedder (2005)*, where the errors of the optimal interpolation formulation are explained in detail.

- We have attempted to add details to better explain how the reconstructions obtained in the lower frequency bands can be used to reconstruct in the following frequency bands (lines 192 - 201).

- In the maps in figure 1, labels with the names of the stations discussed throughout the manuscript have been included.

- Equation (5), which contained 2 typos in the previous version of the manuscript, has been corrected. In addition, more details on the second part of the equation have been added, and a reference has been included: *Pozzi et al 2012* (lines 249 – 250).

- The typo in the caption to figure 5 has been corrected by replacing "serie" by "series".

- It has been specified that what is represented in figure 8 are the average values of the interpolation errors at each point, for the unification of the reconstructed series in the different frequency bands (line 312, and title of figure 8).

- The outdated reference on the estimate of the global rate of sea level rise has been changed, citing *Dangendorf et al. (2017)* in the new version.

- The approach of comparing the correlations obtained by the reconstructions with the main climate indices with the results of *Martínez-Asensio et al. (2014)* has been modified. This comparison was being carried out with the results of the barotropic model by Martinez-Asensio et al. (2014), and now it is done with the result of the tide-gauge data of this work (lines 397 – 405).

- Regarding the ability of the optimal interpolation method to capture small-scale coastal processes, some clarifications on the accuracy of the method have been included in the conclusions (lines 461 - 466).

- The numbering of the figures in the appendices was wrong in the previous version and has been corrected.

- In the data availability section, the persistent identifier DOI of the dataset developed in this work, hosted in the PANGAEA repository, has been added.

- The sections *Competing interest* and *Financial Support* have been added.

- The references have been revised in detail and adapted to the Copernicus style. The main changes compared to the previous version concern the abbreviations of the journal names.

---

## Author Response (AR2)

**Point-by-point reply to Topic Editor**

**C1:** Thank-you for your reply to reviewers. I think you have addressed their points sufficiently on the whole. I just have a couple more comments before it goes for typesetting.

**R1:** We would like to thank you again for handling our manuscript. The answers to your comments are given in the following, and the corresponding changes have been implemented in the new version of the manuscript, which we believe has been improved.

**C2:** One reviewer asked [comment C3] about controlling for the tide in your analysis, and I don't think you've answered this sufficiently. Daily means may not be sufficient to remove the tidal signal, and it is standard practice to use a filter such as Doodson or Demerliac as well as applying means. You need to justify this more carefully. (I suspect it's OK in the Mediterranean because of the small tidal range compared to other variability, but would not be applicable if your technique were applied in other areas. So please check the size of the remaining tidal signal).

**R2:** We would like to clarify that removing the astronomical tide was not our objective. The separation of the sea level anomaly series was carried out by frequency bands, not by physical processes. If we wanted to remove the tides, we would have done it using other methods, as we agree that a daily averaging of time series is not sufficiently accurate.

The aim of the separation in frequency bands was because spatial correlations usually depend on time frequencies, and therefore the interpolation carried out for each separate band is expected to be more accurate than carrying out the interpolation for the whole field.

In any case, the editor is right in that for the Mediterranean tidal range a daily average removes most of the tidal signal. In order to illustrate this, we show some computations carried out for the Ibiza tide gauge record. The RMS difference between daily averaging the original series and daily averaging the de-tided series is of the order of 0.5 cm. The tidal signal was adjusted from the original hourly series of the Ibiza tide gauge, using the UTIDE matlab toolbox (https://es.mathworks.com/matlabcentral/fileexchange/46523-utide-unified-tidal-analysis-and-prediction-functions).

[Figure]

[Figure]

Because tides are not mentioned at all in the manuscript, we would prefer to leave as it is now, in order to make it as simple as possible.

**C3:** Fig 5 annual signal in 1-10yr?

**R3:** Yes, this band shows a clear annual component, quantified in the periodogram showed in the following. The periodogram has been obtained applying a fast Fourier transform to the reconstructed series of the cross-validation test at the closest point to Tarifa tide gauge. The peak corresponds to the annual component of the seasonal cycle of sea level, which lies on the boundary between this band and the band of periods between 1 month and 1 year, and must necessarily fall in one of the two. The use of a Butterworth filter of order 10 aims to achieve an attenuation that is abrupt at the cut-off frequencies, in order to achieve a proper separation between the bands.

[Figure]

**C4:** Line 176 surely just "the trace of matrix []"

**R4:** Sorry, this was a mistake, indeed, but not the one pointed out by the editor. We intended to refer to the g-element of the **diagonal** (not the trace) of matrix [•]. Besides, there was another mistake, this one in formula (4): if matrix $\theta$ is defined as in (2), then the matrix appearing in (4) must be $[\theta\ \mathbb{T}^{-1}\theta^{T}]$ instead of $[\theta^{T}\mathbb{T}^{-1}\theta]$. In summary, lines 175-177 should read:

$$\varepsilon_g = \sigma_g\,(1 - Diag\_g[\theta\ \mathbb{T}^{-1}\theta^{T}]) \tag{4}$$

where $\sigma_g$ is the variance of the signal at point *g*, and $Diag\_g[\bullet]$ denotes the element of the diagonal of matrix $[\bullet]$ corresponding to the interpolation point *g*.

**C5:** Figure 2: Using a scatterplot has the disadvantage that certain spots obscure others. It is particularly a problem in the Gibraltor area in panel d, where there is a big difference between neighbouring points and one is hard to see. You need to find a way of plotting this to avoid this problem. It might be better if you sort the sites along the coast or by longitude before plotting, but maybe just a larger plot will work.

**R5:** We agree that the overlapping between spots makes it difficult to visualise the results. We consider that the best alternative is to move the overlapping spots, linking them with arrows to their location. This is how figure 2 would look like after applying this solution to the Strait of Gibraltar, the French blue coast, the pair Valencia-Sagunto and the pair Ibiza-Formentera. In the new version of the manuscript this way of representing the dots is also used in figures 3, 4 and 12.

[Figure]

**C6:** Figures in the Appendix - ideally, these would be sorted by location along the coast rather than alphabetically, so we can see any patterns that emerge. But consider this optional.

**R6:** Thanks for the suggestion, we have ordered the stations along the coastline in figures A1, A2 and A3.

**List of relevant changes in the manuscript**

- In table 2, some dates in dd/mm/yyyy format have been changed to yyyy-mm-dd format. In particular, the initial dates of the tide gauges at Cagliari, Genoa, Marseille and Porto Maurizio.
- Equation 4 has been corrected, and it has been clarified that this equation refers to the g-element of the diagonal (not the trace) of the matrix.
- Figures 2, 3, 4 and 12 have been modified, separating those points that overlapped with each other and linking them with arrows to their location, in order to facilitate the visualisation of the results.
- In figures A1, A2 and A3 the stations have been rearranged along the coastline. In addition, the names of some stations have been corrected so that now they appear with the same name than in tables 1 and 2.